# Implantable niche with local immunosuppression for islet allotransplantation achieves type 1 diabetes reversal in rats

Jesus Paez-Mayorga [1,2], Jocelyn Nikita Campa-Carranza [1,2], Simone Capuani [1,3], Nathanael Hernandez[1], Hsuan-Chen Liu[1], Corrine Ying Xuan Chua[1], Fernanda Paola Pons-Faudoa [1], Gulsah Malgir[1], Bella Alvarez[1,2], Jean A. Niles[4], Lissenya B. Argueta [4], Kathryn A. Shelton[5], Sarah Kezar[5], Pramod N. Nehete[5,6], Dora M. Berman[7,8], Melissa A. Willman[7], Xian C. Li[9,10], Camillo Ricordi[7], Joan E. Nichols[4,9], A. Osama Gaber[9], Norma S. Kenyon[7,8,11,12,13] & Alessandro Grattoni [1,9,13,14] ✉

Pancreatic islet transplantation efficacy for type 1 diabetes (T1D) management is limited by hypoxia-related graft attrition and need for systemic immunosuppression. To overcome these challenges, we developed the Neovascularized Implantable Cell Homing and Encapsulation (NICHE) device, which integrates direct vascularization for facile mass transfer and localized immunosuppressant delivery for islet rejection prophylaxis. Here, we investigated NICHE efficacy for allogeneic islet transplantation and long-term diabetes reversal in an immunocompetent, male rat model. We demonstrated that allogeneic islets transplanted within pre-vascularized NICHE were engrafted, revascularized, and functional, reverting diabetes in rats for over 150 days. Notably, we confirmed that localized immunosuppression prevented islet rejection without inducing toxicity or systemic immunosuppression. Moreover, for translatability efforts, we showed NICHE biocompatibility and feasibility of deployment as well as short-term allogeneic islet engraftment in an MHC-mismatched nonhuman primate model. In sum, the NICHE holds promise as a viable approach for safe and effective islet transplantation and long-term T1D management.

Clinical pancreatic islet transplantation (CIT) holds promise to transform type 1 diabetes (T1D) management. In CIT, isolated islets are transplanted into the portal circulation and engraft passively in hepatic sinusoids. As pancreatic islets provide dynamic glucose control, CIT significantly improves glycemic profile, decreases hypoglycemic events, and reduces progression of diabetes-related comorbidities[1]. However, hypoxia-related graft attrition and need for systemic immunosuppression limit its widespread clinical adoption. Specifically, systemic immunosuppression is toxic and leaves patients susceptible to life-threatening opportunistic infection and neoplasm development[2,3]. In fact, immune suppression-related adverse effects are detrimental to the point CIT is often reserved for patients already receiving immunosuppression to sustain other transplanted organs.

Pancreatic islet encapsulation promises to overcome the need for toxic systemic immunosuppression regimens[4]. Most encapsulation strategies rely on physical immunoisolation; where islets are enveloped in semipermeable materials that permit oxygen, nutrient, and metabolite exchange, yet limit immune cell infiltration[5,6]. However, impaired diffusion kinetics through the encapsulating material and within the implant compromise long-term cell viability. Moreover,

foreign body reaction (FBR) to the encapsulating material engenders pericapsular fibrotic overgrowth, which further hinders mass transfer and exacerbates graft hypoxia[7]. As a result, conscientious efforts have been made to reduce graft site hypoxia through exogenous oxygen supplementation, in situ oxygen generation, mathematical modeling-based design optimization, and material modification to enhance peri-implant vascularization[8–11]. Although these efforts are promising, the clinical success of physical immunoisolation has been limited.

Pancreatic islets are densely vascularized in their native state and secrete paracrine signals through intra-islet capillary networks[12]. Thus, recent focus has shifted to strategies that permit blood vessel pene-tration through the encapsulating material for direct islet revascularization[13,14]. Apposition of islets to blood vessels grants them a virtually infinite supply of oxygen and nutrients to sustain metabolic capacity and anti-diabetogenic function. However, vascular integra-tion inherently exposes the graft to the immune system, requiring systemic immunosuppression to prevent rejection.

Taken together, there is an unmet need for an encapsulation approach that integrates facile mass transfer to maintain graft function and effective immune evasion to prevent rejection. To meet this need, we developed the Neovascularized Implantable Cell Homing and Encapsulation (NICHE) device[15–17]. The NICHE is a dual reservoir device integrating direct vascularization for oxygen and metabolite exchange and localized immunosuppressant delivery for islet rejection prophy-laxis. Vascularization of the NICHE central cell reservoir is achieved by leveraging the pro-angiogenic properties of a mesenchymal stem cell (MSC) hydrogel, while integration of an interconnected outer drug reservoir permits direct and local immunosuppressant delivery into the cell reservoir through a nanoporous membrane (Fig. 1a).

Here, we demonstrated NICHE efficacy for allogeneic islet trans-plantation and long-term diabetes reversal in an immunocompetent rat model. We investigated the use of two materials, acrylic resin and polyamide (PA) for NICHE manufacturing and demonstrated PA is superior for our approach. We also demonstrated allogeneic islets transplanted within pre-vascularized NICHE were engrafted, revascu-larized, and reverted diabetes in rats for over 150 days. Notably, we confirmed that localized immunosuppression prevented islet rejection without inducing systemic immunosuppression. For translatability efforts, we showed NICHE biocompatibility and feasibility of deploy-ment, as well as short-term allogeneic islet engraftment in an MHC-mismatched nonhuman primate model.

## Results
### NICHE fabrication and characterization
To integrate direct vascularization and local immunosuppression, the NICHE was designed as a dual reservoir platform (Fig. 1a). An external *U*-shaped drug reservoir, where immunosuppressive drugs are loaded, surrounds a central cell reservoir, where islets are transplanted. The cell reservoir is enclosed by a two-layer mesh: an inner 300 μm × 300 μm mesh provides structural support, while an outer 100 μm × 100 μm mesh facilitates vascular tissue penetration and cell retention (Supplementary Fig. 1). The drug and cell reser-voirs are interconnected by 1 × 1 mm openings along each long-itudinal side of the drug reservoir for a net exchange surface area of 8 mm². Moreover, each longitudinal side has nanoporous membranes affixed over the aforementioned openings to permit concentration-driven diffusion of immunosuppressants into the cell reservoir for localized immunosuppression. Finally, biocompatible, self-sealing silicone plugs in the drug and cell reservoirs permit minimally inva-sive, transcutaneous access for immunosuppressant replenishment and islet transplantation.

The NICHE was designed for subcutaneous (SubQ) deployment to minimize procedure invasiveness and was fabricated via additive manufacturing (otherwise known as 3D printing) to facilitate design optimization and scalability. As such, we explored the use of two

biocompatible, 3D printable materials previously employed by our group to fabricate implantable devices: a photopolymeric acrylic resin and PA 2200[17,18]. Firstly, we assessed the tissue reactivity towards subQ implanted resin and PA devices in rats. For comparison, medical-grade titanium (Ti) implants were used as controls. After 6 weeks of implantation, the fibrotic layers around resin and Ti devices were ~50 μm thick and mainly composed of dense collagen (Fig. 1b, d and Supplementary Fig. 2a). In contrast, PA devices had ~2-fold thicker fibrotic layers (~114 μm) that were dense only at the tissue-material interphase (<5 cells in thickness), while lax and vascularized in the remaining portion (Fig. 1c, d). Implant reactivity scoring performed by a blinded, board-certified pathologist showed resin and PA devices had comparable tissue reactivity to Ti control (Fig. 1e; Supplementary Fig. 2b, c). Moreover, both resin and PA had minimal immune cell infiltration and no adjacent tissue necrosis or inflammation (Supple-mentary Fig. 2b), indicating that both materials were biocompatible and well tolerated.

Next, we explored the integration and vascularization of resin and PA NICHE into the subQ space. We previously demonstrated that device implantation with MSCs significantly enhances subQ integra-tion and cell reservoir vascularity[17,19]. As such, we implanted MSC-loaded resin and PA NICHE subQ in rats for 6 weeks. Gross examination of explanted NICHE revealed stark biointegration differences between resin and PA. Of the resin NICHE, only 1 of 4 had tissue ingrowth spanning the entirety of the cell reservoir, indicating full integration within the subQ space; 1 of 4 was mainly filled by clotted blood; and 2 of 4 were completely empty (Fig. 1f). In contrast, 4 of 4 PA devices were fully integrated within the subQ space (Fig. 1g).

Material roughness, hydrophilicity, and stiffness influence cell adhesion and migration, which are key for NICHE subQ integration[20]. Thus, we used atomic force microscopy (AFM), contact angle ana-lysis, and three-point flexural testing to characterize the roughness, hydrophilicity, and stiffness of resin and PA. Resin NICHE had average roughness, contact angle, and elastic modulus of 751.8 ± 181.2 nm, 76.19°, and 3.07 ± 0.14 GPa, respectively. For PA, these values were 1532 ± 657.5 nm, 91.99°, and 1.79 ± 0.17 GPa, respectively (Fig. 1h, i). This data indicated PA had a rougher, more hydrophobic, and less elastic surface than resin, which could explain the differ-ence in device integration. Histological assessment of cell reservoir tissue revealed similar vascular densities between resin (307.6 ves-sels/mm²) and PA (314.2 ± 57.8 vessels / mm²) NICHE (Fig. 1j, k). However, only 1 resin NICHE had sufficient tissue to be processed for histology. Taken together, this data demonstrated that, although both resin and PA were biocompatible and well tolerated, PA was superior for NICHE biointegration and vascularization. As such, all further testing was performed with PA NICHE (hereinafter referred to as NICHE).

### In vitro immunosuppressant release and toxicity
The rate of immunosuppressant delivery with NICHE can be adjusted by modifying the exchange surface area and drug concentration loaded[17]. Clinical islet transplantation requires multi-targeted immu-nosuppression, often employing antibody-based drugs. As such, we characterized the in vitro release of immunosuppressant Cytotoxic T lymphocyte-associated antigen-4-Immunoglobulin (CTLA4Ig), a fusion protein containing the extracellular domain of CTLA4 and human IgG1 (molecular weight: 92 kDa) from NICHE. Additionally, we used IgG (molecular weight: 150 kDa) as a surrogate molecule for full sized antibodies to characterize drug release from NICHE. Congruent with polymer-based release, CTLA4Ig had an initial burst release of ~18% of drug per day in the first 2 days, which steadily decreased and nor-malized to ~1-2% of drug per day by day 12 through day 30 (Fig. 1l). IgG displayed a different release profile with steady release between 5-8% per day for 10 days followed by a steady decrease thereafter, reaching ~0.5% by day 30 (Fig. 1l). These release profiles could be leveraged to

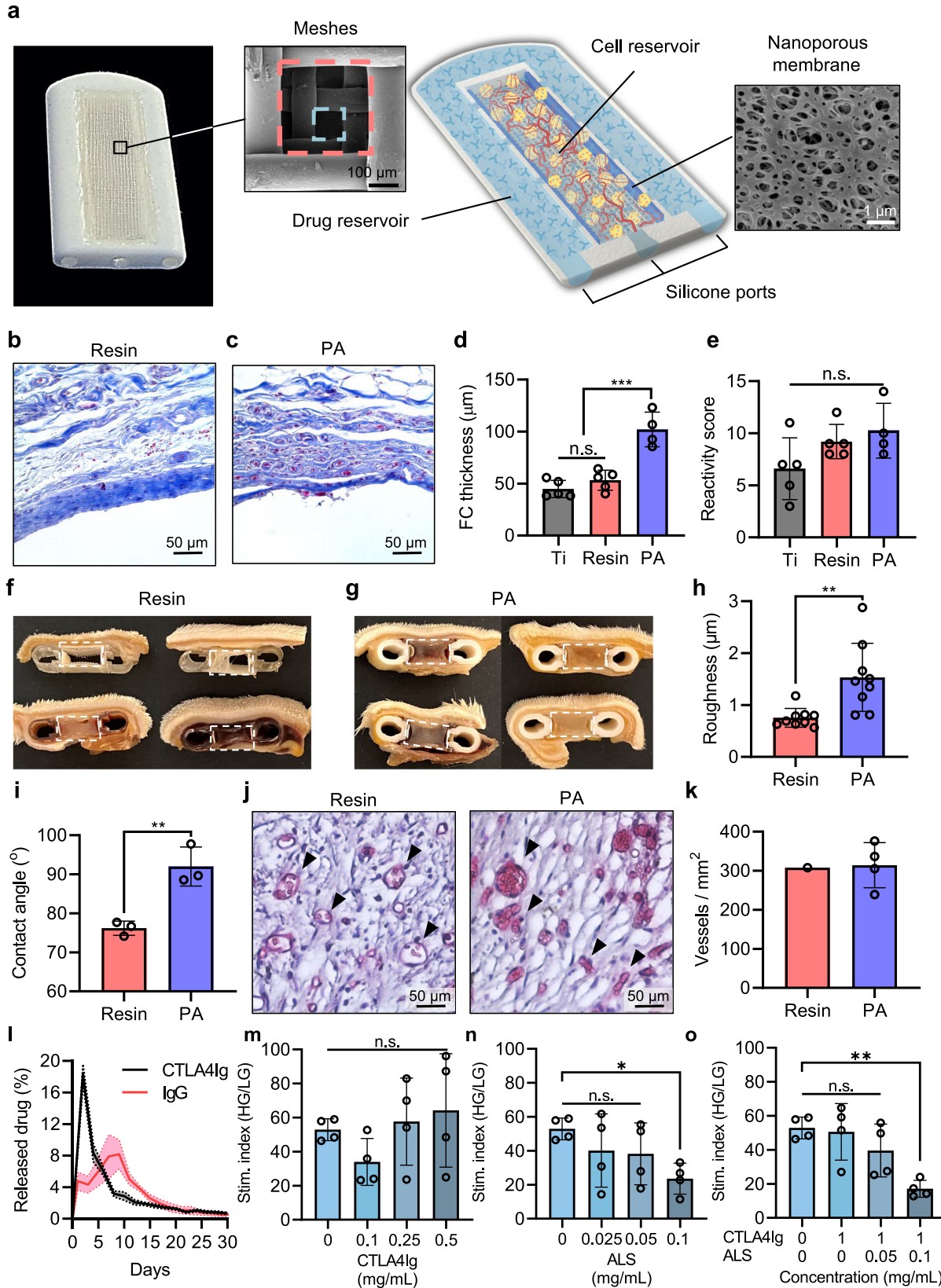

recapitulate the clinical standard of sequential induction and maintenance phases of immunosuppression locally.

Local release of immunosuppressants into the transplant microenvironment entails constant islet exposure to drugs, which could be toxic. Therefore, we investigated the effect of CTLA4Ig and antilymphocyte serum (ALS), two clinically relevant immunosuppressants, on

rat islet function via glucose-stimulated insulin release (GSIR) in vitro (Fig. 1m–o). CTLA4Ig binds to CD80/CD86 receptors on antigen-presenting cells (APCs), blocking co-stimulation signaling and, thus, T-cell activation[21]. ALS is the rat analogue of thymoglobulin (ATG), a polyclonal pan-lymphocyte-depleting antibody highly effective for islet transplantation[22]. Exposure to CTLA4Ig for 5 days did not affect

**Fig. 1 | NICHE design and characterization. a** Optical image of NICHE and annotated rendering of NICHE and scanning electron microscopy (SEM) images of the two-layer mesh and nanoporous membrane (representative scan images of 5 independent samples). The longitudinal section of NICHE is representative for clarity. Representative Masson's trichrome staining of fibrotic capsule around **b** resin and **c** PA devices implanted in rats for 6 weeks. **d** Quantification of fibrotic capsule thickness around medical grade titanium (Ti; $n = 5$), resin ($n = 5$), and PA ($n = 4$) devices implanted subQ for 6 weeks in rats, mean ± SD, one-way analysis of variance (ANOVA) ***$p < 0.001$. **e** Implant reactivity scores of Ti ($n = 5$), resin ($n = 5$), and PA ($n = 4$) devices implanted subQ for 6 weeks in rats, mean ± SD, one-way ANOVA, n.s. $p = 0.1117$. Optical images showing cross sections of **f** resin and **g** PA NICHE implanted subQ for 6 weeks. Dashed lines indicate cell reservoir. **h** Resin and PA roughness quantified using atomic force microscopy ($n = 9$ devices per material), mean ± SD, unpaired two-tailed student's t-test **$p < 0.01$. **i** Contact angle analysis of resin and PA devices ($n = 3$ devices per material), mean ± SD, unpaired two-tailed student's t-test ***$p < 0.001$. **j** Resin and PA NICHE cell reservoir tissue sections stained with blood vessel marker *B. simplicifolia* lectin; vessels in red pointed by black arrow heads. **k** Vascular density quantification of resin ($n = 1$) and PA ($n = 4$) cell reservoir tissue from independent NICHE, mean ± SD. **l** Daily percent in vitro release of CTLA4Ig and IgG from NICHE drug reservoir ($n = 4$ devices per drug), mean ± SD. Static glucose stimulated insulin release (GSIR) of rat islets after 5 days of incubation with **m** CTLA4Ig, **n** ALS, and **o** CTLA4Ig + ALS ($n = 4$ per condition), mean ± SD, one-way ANOVA, *$p < 0.05$, **$p < 0.001$, n.s. $p > 0.05$ of each condition versus vehicle (0 mg/mL). Source data are provided as a Source Data file.

islet function at any dose tested up to 1 mg/mL (Fig. 1m, o). Exposure to ALS showed a negative, albeit not statistically significant, effect on islet function at 0.1 mg/mL, which was set as the toxicity threshold (Fig. 1n). Moreover, exposure to a combination of 1 mg/mL CTLA4Ig and up to 0.05 mg/mL ALS did not exacerbate toxicity, indicating immunosuppressant co-delivery was safe for islets (Fig. 1o).

### Effect of local immunosuppressant release on angiogenesis

Vascular integrity within NICHE is key for graft vascularization. Therefore, we assessed the effect of local immunosuppressant release on angiogenesis. We found that low- and high-dose CTLA4Ig, ATG and CTLA4Ig + ATG did not affect tube formation capacity of human endothelial cells in terms of tube length or number of segments (Fig. 2a–c). Furthermore, immunosuppressant exposure did not affect the expression of angiogenic genes *cdh5, nos3*, and *vegf* compared to vehicle controls (Fig. 2d). High-dose, but not low-dose, ATG induced a 4-fold increase in *vcam* expression, a gene associated with endothelial inflammation[23].

We further characterized the effect of these immunosuppressants on angiogenesis in vivo. Rats were implanted subQ with 1 NICHE on each flank and allowed to vascularize for 6 weeks. Next, one of the two NICHE was transQ filled with CTLA4Ig + ALS for local release, while the contralateral NICHE was filled with saline and served as a control. Two weeks later, NICHE were harvested and assessed for angiogenic signaling and vascularization. Quantification of VEGF levels served as a surrogate of angiogenic signaling while immunohistochemical labeling with CD31, VE-Cadherin and eNOS informed on vessel maturity and function. In line with in vitro gene expression analysis, VEGF levels were similar in the microenvironment of control and locally immunosuppressed NICHE (Fig. 2e). Additionally, immunohistochemical analysis showed CD31, VE-Cadherin, and eNOS expression was comparable between control and locally immunosuppressed NICHE ($p > 0.5$ across groups), indicating that NICHE vasculature was mature and functional independently of local CTLA4Ig-ALS release (Fig. 2f, g). Taken together, this data indicated that CTLA4Ig and ATG/ALS release did not affect angiogenic signaling, vessel maturity, or function.

### NICHE efficacy testing for allogeneic islet transplantation and diabetes reversal

We assessed NICHE efficacy for transplantation of allogeneic islets to treat diabetes in an immunocompetent rat model using Lewis rats as donors and Fischer 344 rats as recipients (Fig. 3a). NICHE loaded with syngeneic MSCs were subQ implanted in rats and allowed to vascularize for 6 weeks. At 2 weeks post-implantation, rats were rendered diabetic via streptozotocin injection. After pre-vascularization, diabetic rats were randomized into three immunosuppressive regimens (day 0): "NICHE" in which CTLA4Ig and ALS were loaded into the NICHE drug reservoir for local delivery; "IP", which received a clinically relevant regimen consisting of induction with ALS + CTLA4Ig, followed by maintenance with CTLA4Ig alone; and "No IS", which did not receive any immunosuppression. Three days later, leveraging the immunomodulatory and graft-promoting properties of MSCs, allogeneic islets were co-transplanted with syngeneic MSCs into the NICHE cell reservoir of all rats (Supplementary Fig. 3).

The isolated islets had typical morphology and responded to glucose stimulation in vitro prior to transplantation (Fig. 3b, c). By day 28 after islet transplantation, non-fasting blood glucose (BG) levels had dropped in NICHE and IP rats to 375.07 ± 117.68 mg/dL and 399.84 ± 124.36 mg/dL, respectively, whereas No IS control rats had significantly higher BG levels of 520.43 ± 24.39 mg/dL. To potentiate therapeutic effect, a second islet transplantation was performed on day 31 in rats from all groups. Within 10 days, 66.7% NICHE (8/12 animals) and 62.5% of IP (5/8 animals) became euglycemic (BG < 200 mg/dL). Non-euglycemic rats (4 of 12 NICHE, 3 of 8 IP, and 4 of 4 No IS) were euthanized on day 63 as humane endpoint. Two rats in the NICHE group rejected their grafts prematurely on day 84 due to immunosuppressant release failure and were removed from study. One euglycemic rat in the NICHE group was removed day 90 due to injury. On day 115, NICHE was explanted from remaining IP rats and 2 of 5 NICHE rats, while the remaining 3 NICHE rats were followed up long term and explanted on day 193. Upon NICHE explantation, all rats immediately reverted to hyperglycemia, confirming diabetes correction was due to the transplanted islets that were retained within the NICHE. Moreover, to assess long-term glycemic control, HbA1c was quantified at the study endpoint. NICHE and IP rats had HbA1c comparable to healthy animals, below that of the desired threshold for diabetic patients of 7%[24] (Fig. 3e). In contrast, No IS rats had HbA1c > 20%, indicative of extreme and unmanaged diabetes. It is noteworthy that the NICHE and IP rats that did not become euglycemic (~33%) maintained a significant improvement in BG levels compared to No IS controls until endpoint (Supplementary Fig. 4), suggesting that the engrafted marginal islet mass had not been rejected.

Islet function of euglycemic NICHE and IP rats was assessed via intraperitoneal glucose tolerance test (ipGTT) on day 115 and compared to healthy and diabetic controls. NICHE and IP rats responded to glucose challenge similarly to healthy animals and significantly better than diabetic controls (Fig.3f, g). Likewise, NICHE and IP rats had fasting c-peptide levels that were comparable to healthy rats and significantly higher than diabetic controls, further demonstrating graft functionality (Fig. 3h). Weight fold change was used as an indicator of overall animal well-being. Upon diabetes induction, weight gain stalled in all animals (Fig. 3i). However, after islet transplantation, NICHE and IP rats had an immediate improvement in weight gain, while No IS rats remained with ill-thrift through endpoint, underscoring that the therapeutic benefit was exclusive to NICHE and IP groups.

Islets are densely vascularized in their native microenvironment; for transplantation, effective revascularization is imperative to achieve successful engraftment and therapeutic function. As such, we quantified the percent islet area covered by blood vessels (Fig. 3j–m). Islets in the NICHE and IP groups averaged 12.02 ± 2.06% and 10.66 ± 2.41% vessel area, which was significantly higher than that of islets in the native pancreas (7.58 ± 1.63%), indicating the NICHE microenvironment was conducive to islet engraftment and efficient re-vascularization.

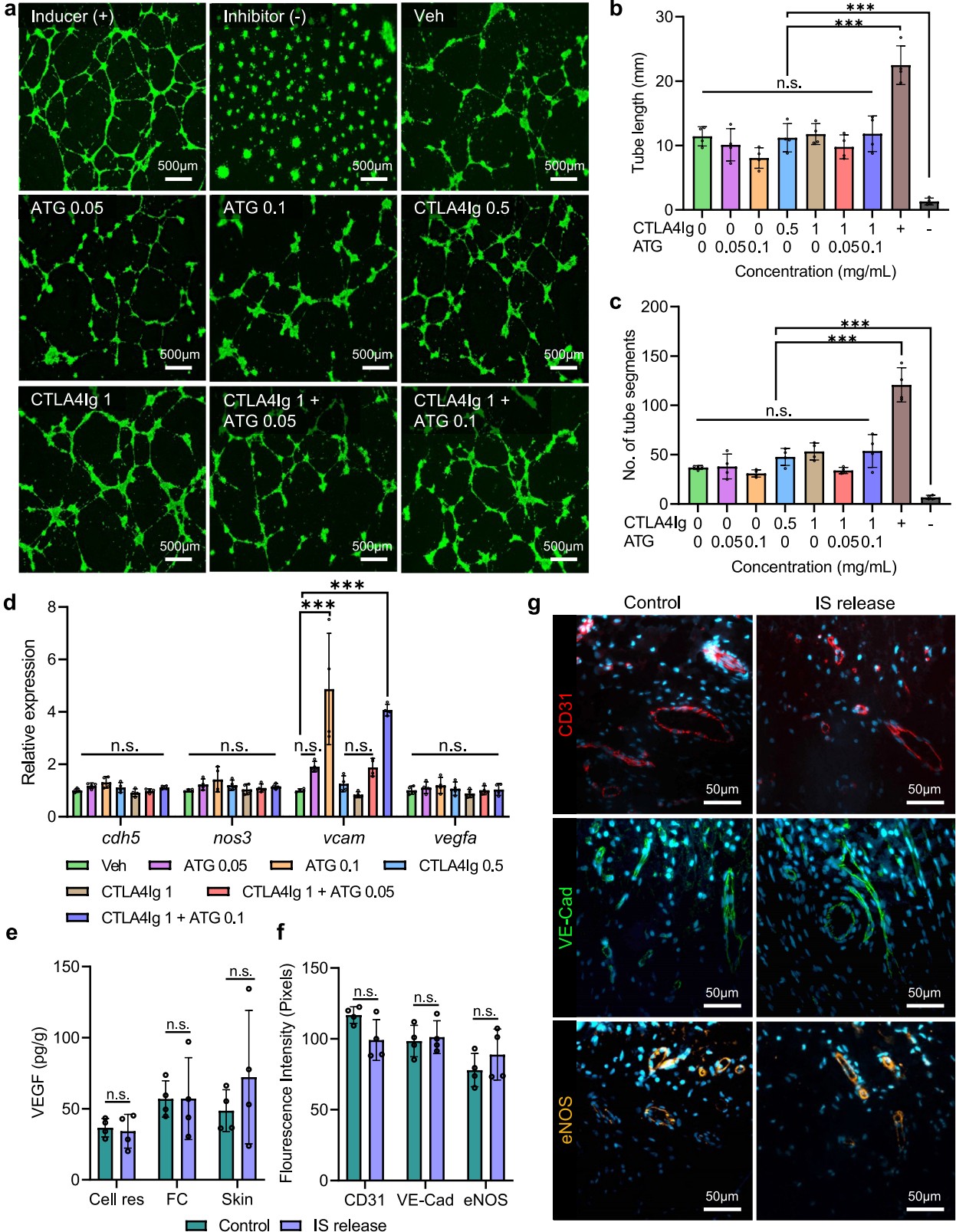

Taken together, we conclude that: (1) the pre-vascularized NICHE microenvironment was conducive to islet engraftment, (2) immunosuppression was needed to observe therapeutic benefit, and (3) local and systemic immunosuppression were equally effective in preventing graft rejection and preserving islet function.

**Local immunomodulation of the transplant microenvironment**

The immunomodulation of the transplant microenvironment by local and systemic immunosuppression was characterized via imaging mass cytometry (IMC) analysis of cell reservoir tissue sections from euglycemic NICHE and IP rats as well as from rats with actively rejecting

**Fig. 2 | Effect of local immunosuppression on angiogenesis. a** In vitro tube formation activity of human umbilical vein endothelial cells (HUVEC) in the presence of different drug concentrations (mg/mL). Supplemented media was used as positive inducer (+), 30 μM suramin as positive inhibitor (-), and non-supplemented media as control (Veh). **b** Total length quantification and **c** number of segments of tube formation ($n = 4$ biological replicates per condition), mean ± SD, one-way ANOVA with Tukey's multiple comparisons test (***$p < 0.001$). **d** qPCR analysis of vascular related genes in HUVEC after treatment with different concentrations of ATG and CTLA4Ig (mg/mL). *Gapdh* gene was used as internal control and fold-change in gene expression is relative to control untreated cells. ($n = 4$ biological replicates), mean ± SD, one-way ANOVA with Tukey's multiple comparisons test (***$p < 0.001$; n.s. $p > 0.05$). **e** Quantification of VEGF levels in cell reservoir and surrounding microenvironment for control and IS release samples ($n = 4$ biological replicates), mean ± SD, unpaired two-tailed student's $t$-test n.s. $p > 0.05$. **f** Fluorescence intensity measurements of IHC analysis in **g** ($n = 4$ biological replicates), mean ± SD, unpaired two-tailed student's $t$-test n.s. $p > 0.05$. **g** Representative sections of NICHE control and IS release slides stained with functional blood vessel markers CD31 (red), VE-cadherin (green), and eNOS (gold). Source data are provided as a Source Data file.

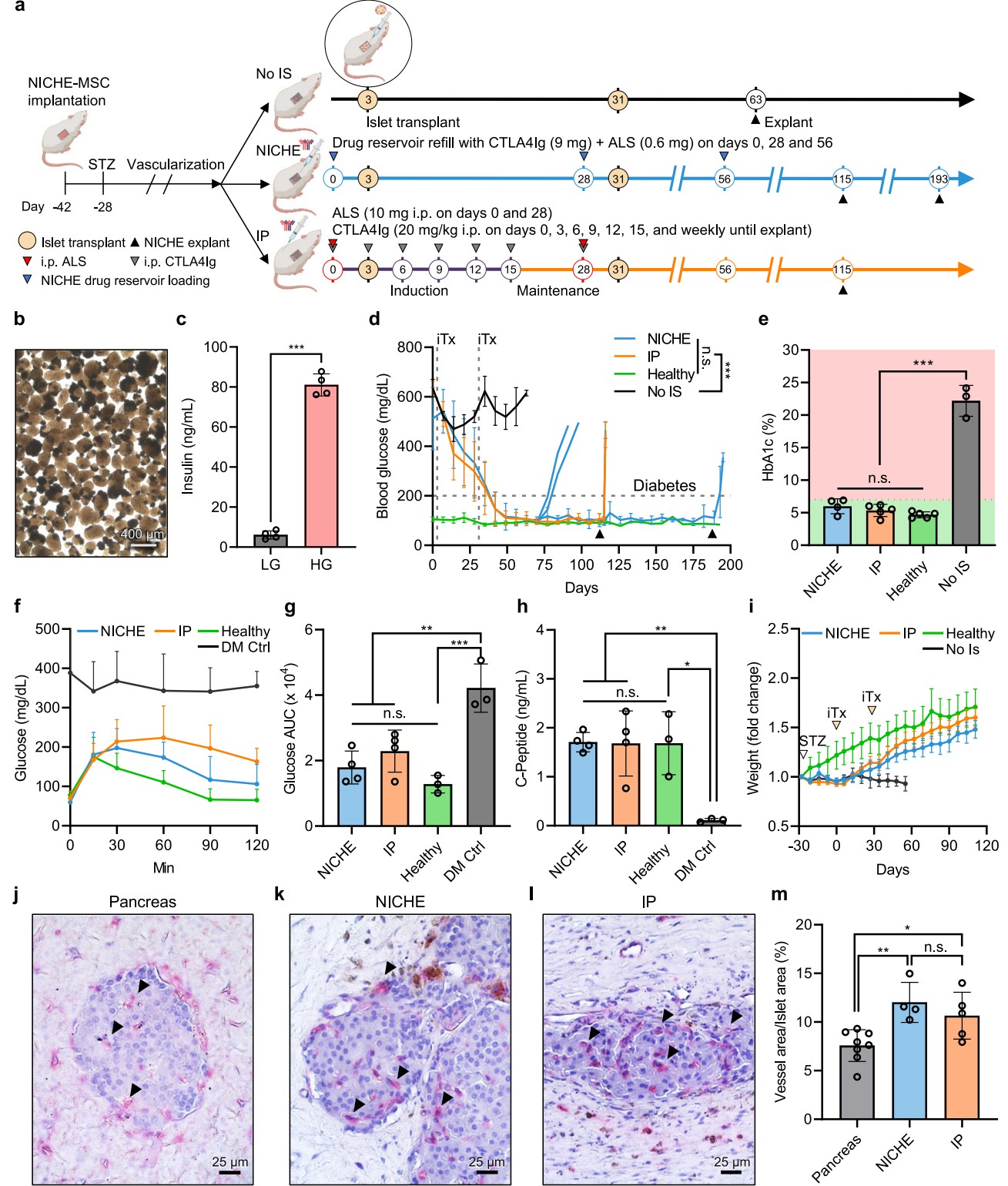

**Fig. 3 | NICHE efficacy testing for allogeneic islet transplantation in immuno-competent diabetic model. a** Study design. STZ streptozotocin; **b** Rat islets pre-transplantation (aliquoted sample from main islet pool). **c** Glucose stimulated insulin release of islets pre-transplantation ($n = 4$), unpaired two-tailed student's $t$-test ***$p < 0.001$. **d** BG measurements of diabetic rats transplanted with islets in NICHE cell reservoir receiving local (NICHE; $n = 8$ to day 84, $n = 7$ to day 91, $n = 5$ to day 116, $n = 3$ to day 195) or systemic (IP; $n = 5$ to day 116) immunosuppression, no immunosuppression (No IS; $n = 4$ to day 63), and healthy controls ($n = 6$ to day 116, $n = 1$ to day 195), iTx = islet transplant. Only NICHE and IP rats that achieved euglycemia are plotted. **e** HbA1c of NICHE ($n = 4$), IP ($n = 5$), healthy ($n = 5$), and No IS ($n = 3$) rats at endpoint. Dotted line indicates glycemic control threshold. **f** BG curves and **g** area under the curve (AUC) of intraperitoneal glucose tolerance test on day 112 of NICHE ($n = 4$), IP ($n = 4$), healthy ($n = 3$), and diabetic control ($n = 3$) rats. **h** Fasting plasma C-peptide on day 112 of NICHE ($n = 4$), IP ($n = 4$), healthy ($n = 3$), and diabetic control ($n = 3$) rats. **i** Weight fold change to baseline of NICHE ($n = 8$), IP ($n = 5$), Healthy ($n = 5$), and No IS ($n = 4$) rats. **j** Pancreas of healthy rats, and cell reservoir tissue of euglycemic **k** NICHE and **l** IP rats stained for blood vessels (red); black arrowheads = capillaries. **m** Percent islet area comprised of vessels in healthy rat pancreas ($n = 8$), NICHE ($n = 4$), and IP rats ($n = 5$). All panels presented as mean ± SD, one-way ANOVA with Tukey's multiple comparisons test (***$p < 0.001$; **$p < 0.01$; *$p < 0.05$; n.s. $p > 0.05$). Source data are provided as a Source Data file.

grafts. The latter were defined as transplanted, euglycemic rats that had spontaneous reversal to diabetic state and histological evidence of active rejection (Supplementary Fig. 5). Non-supervised cluster analysis of IMC samples classified cell populations into 13 clusters, the most predominant of which were: beta cells (insulin[+]), alpha cells (glucagon[+]), vascular cells (αSMA[+]), cytotoxic T cells (CD8[+]), and macrophages (CD68[+]). Alpha and beta cells were predominantly found in NICHE and IP sections and their distribution was consistent with islet morphology (Fig. 4a, b). In contrast, rejecting grafts showed minimal presence of alpha and beta cells, which were scattered and lacked typical islet morphology, suggestive of graft destruction (Fig. 4c, d). Moreover, blood vessels were present across all groups, with clear islet revascularization in NICHE and IP grafts. Rejecting grafts were densely infiltrated with cytotoxic T-cells and macrophages, whereas NICHE and IP grafts had significantly reduced number of these cell types (Fig. 4a–d). Infiltration of helper T-cells was limited across groups; however, higher levels were observed in rejecting grafts (Fig. 4d). The remaining cell population clusters consisted of various inflammatory cell types that were predominantly present in rejecting grafts (Fig. 4e–h). Of note, histological reconstructions and tSNE plots showed homologous cell distribution in NICHE and IP groups, underscoring local and systemic immunosuppression were equally effective in preserving islet viability.

## Memory T-cell immunomodulation with local and systemic immunosuppression

To evaluate the systemic immunomodulatory effect of local and systemic immunosuppression, effector memory T-cell populations were assessed at endpoint in peripheral blood and spleen of NICHE and IP rats with viable grafts, as well as of rats with actively rejecting or previously rejected grafts. Rats with previously rejected grafts were defined as rats from the No IS cohort that never achieved euglycemia and whose histological assessment showed cell reservoirs that had been cleared of islets with minimal inflammation (Supplementary Fig. 5).

In the blood, CD4 and CD8 effector memory cells (Tem) frequencies were similar across NICHE, IP, and previously rejected rats, while rats with active rejection had significantly higher Tem levels (Fig. 4i, j). Blood CD4 and CD8 central memory cells (Tcm) did not differ statistically across groups; albeit a slight trend of increased CD8 Tcm cells was observed in rats with active rejection (Fig. 4i, j). In the spleen, similarly to the blood, CD4 Tem frequencies were similar between NICHE and IP rats (Fig. 4k). However, rats with past rejection showed higher splenic CD4 Tem frequencies (Fig. 4k). Similarly, NICHE and IP groups had comparable CD8 Tem and Tcm frequencies, which tended to be lower than in rats with past rejection (Fig. 4l; $p = 0.0913$ NICHE vs. past rejection; $p = 0.1645$ IP vs. past rejection) and active rejection (Fig. 4l; $p = 0.0529$ NICHE vs. active rejection; $p = 0.0954$ IP vs. active rejection).

Overall, NICHE and IP rats had similar frequencies across all memory T cells in blood and spleen that were consistently lower than in rats with active rejection. Meanwhile, rats with past rejection had increased memory T cell frequencies in the spleen and limited presence in the blood, suggesting homing rather than active circulation of these cell types. Taken together, this data indicated that local immunosuppression with NICHE attenuated memory T cell responses similarly to standard systemic delivery, effectively prolonging graft viability.

## Drug biodistribution and systemic immunosuppression

Exposure to immunosuppressants was assessed via quantification of ALS and CTLA4Ig in plasma, peripheral organs, and the transplant microenvironment. In NICHE rats, ALS (0.6 mg) and CTLA4Ig (9 mg) were loaded into the drug reservoir on days 0, 28, and 56 for sustained release over time (Fig. 3a). In contrast, to recapitulate a clinical scenario, IP rats received induction with combination of ALS (10 mg i.p. on day 0) and CTLA4Ig (20 mg/kg i.p. on days 0, 3, 9, 12, 15), followed by maintenance with CTLA4Ig alone (20 mg/kg i.p. weekly) (Fig. 3a). In preparation for re-transplantation on day 31, IP rats received an additional ALS dose on day 28. Moreover, after observation of graft destabilization (Supplementary Fig. 6), NICHE rats received a "rescue" drug reservoir refilling with ALS and CTLA4Ig on day 157.

On day 6, IP rats averaged plasma ALS and CTLA4Ig levels of 358.18 μg/mL and 150.02 μg/mL, respectively (Fig. 5a, b). ALS levels dropped to 84.87 μg/mL by day 28 and, following readministration, increased to 188.70 μg/mL by day 35. Thereafter, plasma ALS steadily decreased, reaching 5.18 μg/mL at endpoint on day 112. On the other hand, CTLA4Ig levels in IP rats increased during induction, peaking at 246.59 μg/mL on day 14. During the maintenance phase, CTLA4Ig levels normalized to ~104.62 μg/mL through the remainder of the study.

Throughout the study, NICHE rats had ~36- and ~20-fold lower plasma ALS and CTLA4Ig levels than IP rats. On day 6, NICHE rats had circulating ALS levels of 1.33 μg/mL, which increased to 6.30 μg/mL by day 9 and remained steady through day 70 (Fig. 5a). Thereafter, plasma ALS levels steadily decreased and dropped below the limit of quantitation of 0.1 μg/mL after day 143. In accordance with in vitro release profile, CTLA4Ig exhibited a burst release, with peak plasma concentrations of 21.11 μg/mL on day 6 that steadily dropped to 8.21 μg/mL by day 28 (-2.5-fold) (Fig. 5b). On day 35, 7 days after drug reservoir refilling, CTLA4Ig levels rose to concentrations comparable to day 6 (18.56 μg/mL), indicating successful device refilling. Similarly, circulating levels steadily decreased, reaching 6.25 μg/mL on day 56, which was comparable to levels on day 28. Replenishment on day 56 restored CTLA4Ig concentrations within 7 days which, congruent with previous refilling procedures, decreased ~2.3-fold by day 84 (28 days post-refill). Drug reservoirs were not refilled after day 56 to assess how long drug loading could be deferred before islet function was compromised. Because of this, plasma CTLA4Ig levels continued to steadily decrease, reaching 0.02 μg/mL on day 157. Finally, after rescue refilling on day 157, circulating CTLA4Ig levels surged as expected, peaking at ~1 μg/mL on day 164.

Similarly to plasma, IP rats had significantly higher CTLA4Ig accumulation in peripheral tissues than NICHE rats, with 21-fold, 16-fold, 7-fold, and 22-fold higher levels in the kidney, liver, spleen, and lymph nodes, respectively (Fig. 5c). ALS levels in peripheral tissues were below the limit of quantitation in both NICHE and IP rats at

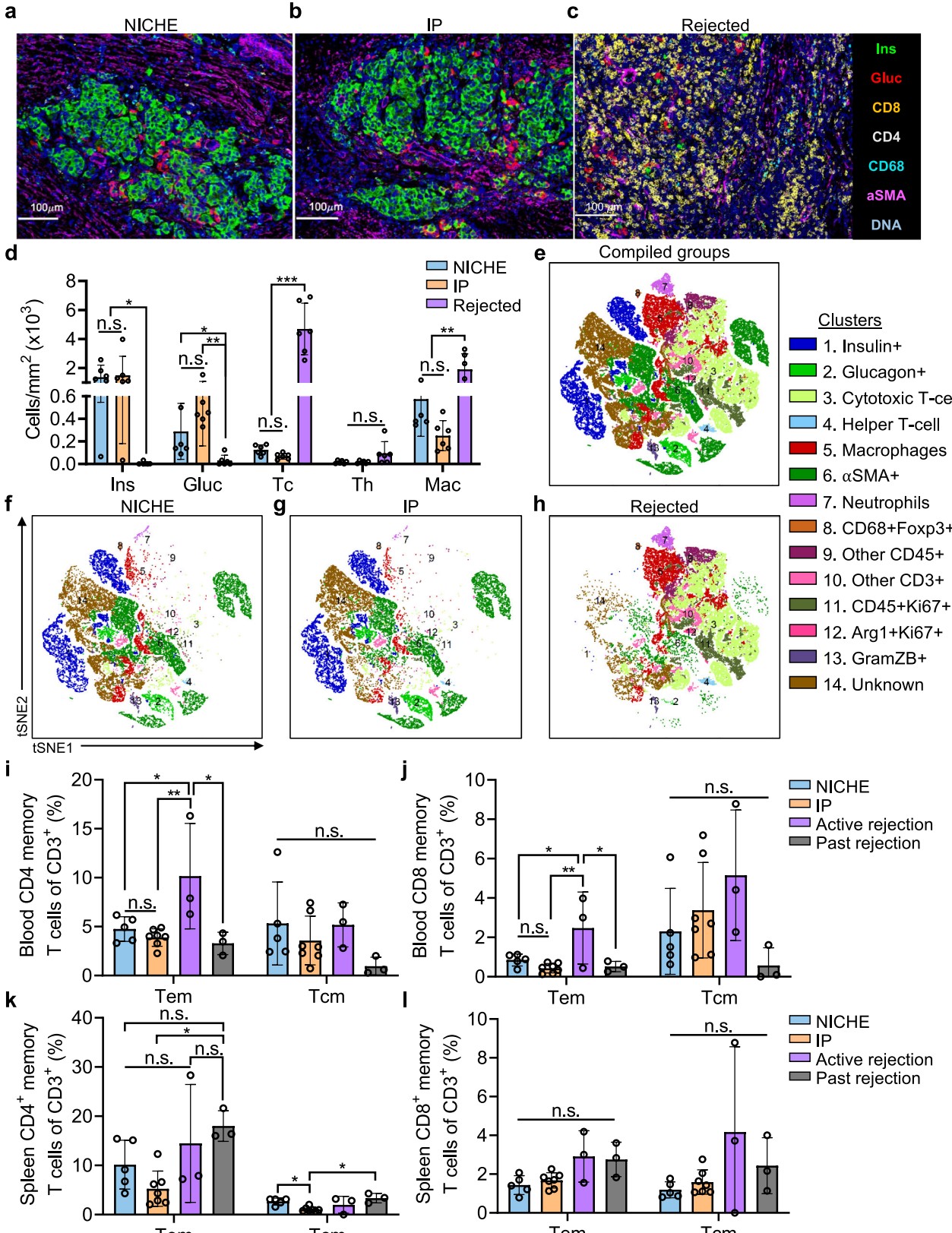

**Fig. 4 | Immunomodulation with NICHE and IP delivery.** Imaging mass cytometry (IMC) of cell reservoir tissues from **a** NICHE, **b** IP, and **c** rats with rejected grafts. **d** IMC cell populations quantification ($n = 6$), mean ± SD, one-way ANOVA with Tukey's multiple comparisons test per population (*$p < 0.05$; **$p < 0.01$; ***$p < 0.001$; n.s. $p > 0.05$). **e–h** tSNE plots of IMC data. Pooled data from $n = 6$ ROI per group. Cell populations were defined as follow: Cytotoxic T-Cell (Tc) CD45+CD3+CD8+, Helper T-cell (Th) CD45+CD3+CD4+, Macrophages (Mac)

CD45+CD68+, Neutrophils CD45+MPO+. Flow cytometry analysis of **i** blood CD4+ memory T cells, **j** blood CD8+ memory T cells, **k** spleen CD4 + memory T cells, and **l** spleen CD8+ memory T cells at endpoint in NICHE ($n = 5$) and IP ($n = 7$) rats, and grafts with active ($n = 3$) or past ($n = 3$) rejection, mean ± SD, one-way ANOVA with Tukey's multiple comparisons test per population (*$p < 0.05$; **$p < 0.01$; n.s. $p > 0.05$). Source data are provided as a Source Data file.

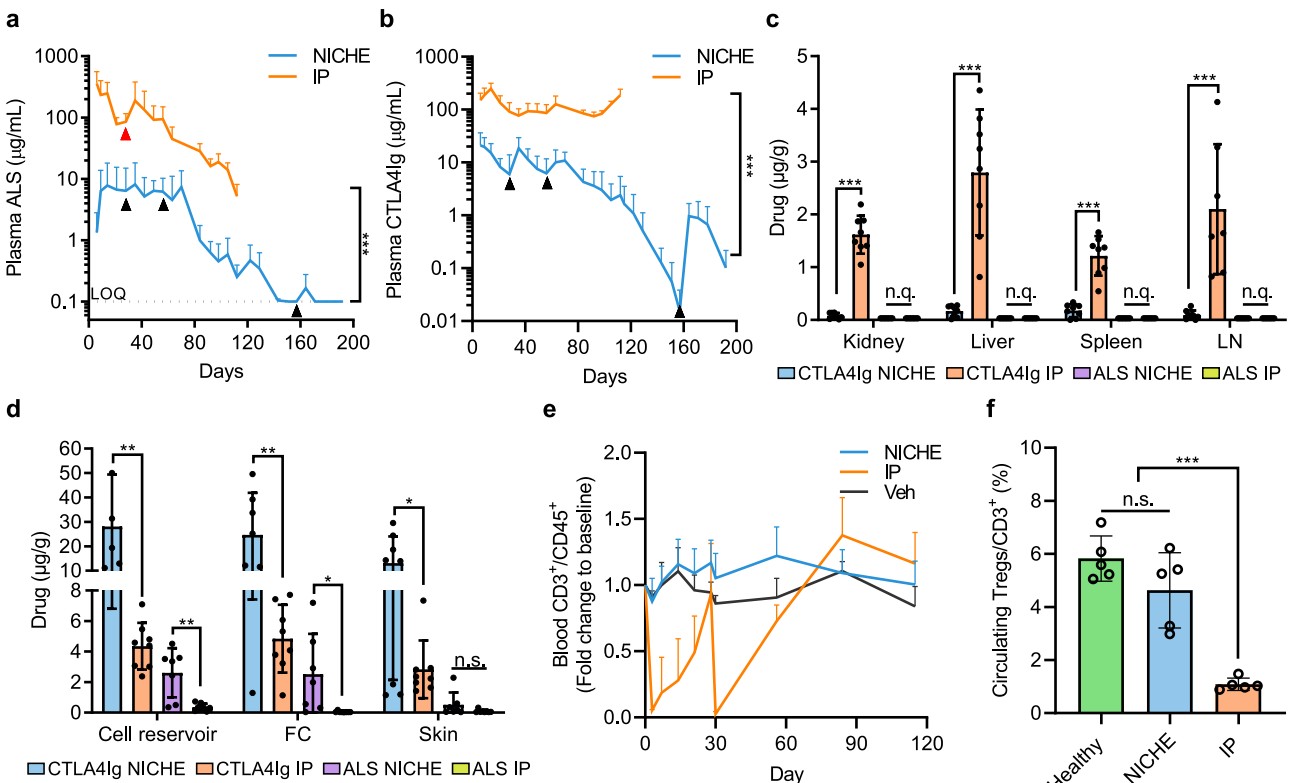

**Fig. 5 | Drug biodistribution and systemic immunosuppression.** Plasma levels of **a** ALS and **b** CTLA4Ig of NICHE ($n = 12$ to day 63, $n = 8$ to day 92, $n = 5$ to day 112, $n = 3$ to day 192) and IP ($n = 8$ to day 63, $n = 5$ to day 112) rats, mean ± SD, unpaired two-tailed student's $t$-test between NICHE and IP overall average concentrations, (***$p < 0.001$). Black triangles indicate drug reservoir reloading. The red triangle indicates systemic ALS bolus. LOQ limit of quantitation. Quantification of immunosuppressants in **c** peripheral organs and **d** transplant microenvironment at endpoint of NICHE and IP rats ($n = 8$), mean ± SD, unpaired two-tailed student's $t$-

test between NICHE and IP rats for each organ (*$p < 0.05$; **$p < 0.01$; ***$p < 0.001$; n.s. $p > 0.05$; n.q. = not quantifiable). FC = fibrotic capsule. **e** Quantification of lymphocytes in blood expressed as fold change to day 0 of NICHE ($n = 8$ to day 56, $n = 5$ to day 115), IP ($n = 5$), and rats receiving vehicle ($n = 4$). **f** Quantification of Tregs in blood on day 84 of NICHE, IP, and healthy rats ($n = 5$), mean ± SD, one-way ANOVA with Tukey's multiple comparisons test (***$p < 0.001$; n.s. $p = 0.1643$). Source data are provided as a Source Data file.

endpoint (Fig. 5c). This was expected as NICHE rats were releasing low dose ALS and IP rats had received their last dose 84 days prior to endpoint (Fig. 3a). At the transplant microenvironment, NICHE rats had significantly higher CTLA4Ig and ALS levels in the cell reservoir, fibrotic capsule, and adjacent skin than IP rats (Fig. 5d). Taken together, this data indicated that localized dosing with NICHE enriched drug at the transplant microenvironment and significantly reduced drug accumulation in plasma and peripheral organs, decreasing the potential for systemic toxicity.

In addition to drug biodistribution, we assessed the degree of immunosuppression resulting from local and systemic delivery. Systemic immunosuppression with ALS causes lymphocyte depletion[25,26], while systemic CTLA4Ig is associated to decreased Treg frequencies[27]. Thus, we quantified circulating T-cells and Tregs in NICHE and IP rats. Following systemic ALS administration on days 0 and 28, IP rats exhibited acute lymphocyte depletion, indicative of profound immunosuppression (Fig. 5e). In contrast, circulating T-cell levels remained unchanged in NICHE rats throughout the study (Fig. 5e). Moreover, Treg frequencies in NICHE rats were similar to healthy controls, whereas IP rats had significantly lower levels (Fig. 5f). It is noteworthy that Treg quantification was performed on day 84, once total T-cell levels had been restored in IP rats, underscoring the immunosuppressive effect of systemic dosing (Fig. 5e). Taken together, this data indicated that the limited circulating levels in plasma resulting from local drug delivery with NICHE did not induce systemic effects and that, in contrast to the IP cohort, NICHE rats were not overtly immunosuppressed.

## NICHE biocompatibility, islet engraftment, and local immunomodulation in NHP

Aiming at translation, we characterized the biocompatibility, islet engraftment, and local immunomodulation of NICHE in nonhuman primates (NHP). Healthy cynomolgus macaques ($n = 6$) were subQ implanted with NICHE loaded with autologous MSCs to promote vascularization (Fig. 6a). Implanted NICHE left a minimal imprint on the macaque skin and no clinical signs of irritation or swelling were observed, indicating the implants were well tolerated (Fig. 6b). Additionally, transcutaneous drug reservoir loading and islet transplantation procedures were minimally invasive (Fig. 6c, d). Finally, NICHE explantation with intact fibrotic capsule was straightforward and required incisions smaller than 2 cm (Fig. 6e).

NICHE from 2 NHP were each explanted after 4 and 6 weeks of implantation to characterize integration in the subQ tissue, reactivity to the implant, and vascularization. By 4 weeks post-implantation, NICHE was well integrated into the subQ space with tissue spanning the entirety of the cell reservoir (Fig. 6f, g). The fibrotic capsule that formed around the implant was ~100 μm, collagenous, and vascularized (Fig. 6h). Moreover, the NICHE had a lower reactivity score than control medical grade Ti implants, underscoring implant tolerability and biocompatibility (Fig. 6i). Labeling of cell reservoir tissue sections with blood vessel marker CD31 confirmed NICHE tissue was extensively vascularized (Fig. 6j). At 4 weeks post implantation, NICHE had ~137 ± 56 vessels/mm² that increased to ~207 ± 55 vessels/mm² by week 6, which accounted for ~8.6% of tissue area occupied by blood vessels (Fig. 6k, l).

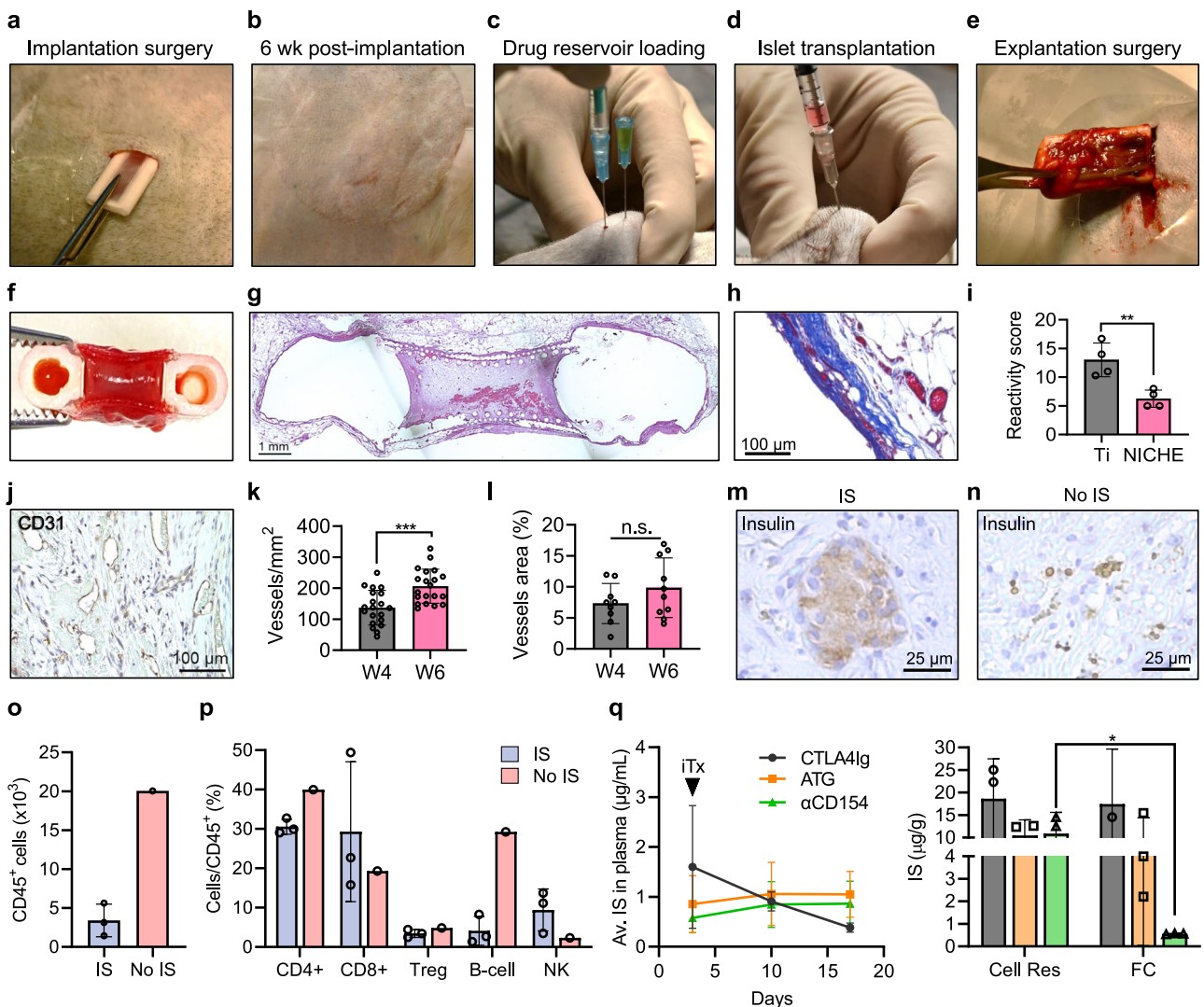

**Fig. 6 | NICHE biocompatibility, islet engraftment, and local immunomodulation in NHP. a–e** NICHE deployment strategy in nonhuman primates (NHPs). **f** Gross and **g** H&E-stained micrograph showing a cross-section of NICHE after 4 weeks of subcutaneous implantation in NHP. **h** Masson's Trichrome staining of fibrotic capsule formed around NICHE after 4 weeks of subQ implantation in NHP. **i** Implant reactivity scores of Ti (*n* = 4) and PA (*n* = 4) devices implanted subQ in NHP, mean ± SD, unpaired two-tailed student's t-test, (**\*\****p* < 0.01). **j** Micrograph of NICHE cell reservoir tissue stained with blood vessel marker CD31. **k** Vascular density quantification of NICHE implanted subQ for 4 and 6 weeks (*n* = 20 ROI), mean ± SD, unpaired two-tailed student's t-test, \*\*\**p* < 0.001. **l** Quantification of percent area of NICHE tissue occupied by blood vessels after 4 and 6 weeks of implantation (*n* = 20 ROI), mean ± SD, unpaired two-tailed student's t-test (n.s. *p* = 0.1988). **m, n** Micrograph of NICHE tissue stained for insulin 14 days post-transplant in IS and No IS NHPs. **o, p** Quantification of cell inside NICHE of IS (*n* = 3) and No IS (*n* = 1) NHPs 14 days post-transplant, mean ± SD. **q** Quantification of immunosuppressants in plasma and NICHE microenvironment of IS NHP (*n* = 3), mean ± SD. Cell Res cell reservoir, FC fibrotic capsule. Source data are provided as a Source Data file.

After 6 weeks of prevascularization, NICHE from 3 of 4 NHP were transcutaneously filled with an immunosuppression cocktail (IS) consisting of anti-thymocyte globulin (ATG), CTLA4Ig, and anti-CD154 (αCD154) for local delivery (Fig. 6c); 1 of 4 NHP did not receive IS loading and served as a negative control. αCD154 blocks the CD40-CD154 co-stimulatory pathway, halting pro-inflammatory cytokine production, effector T-cell activation, and B-cell class switch, while promoting T-reg expansion[28]. Three days later, allogeneic islets from MHC mismatched cynomolgus macaques were transcutaneously co-transplanted with autologous MSCs into the cell reservoir of all NHP (Fig. 6d). NICHE were harvested from all animals 14 days post-transplantation as allograft rejection typically occurs within this timeframe in the absence of immunosuppression, thus permitting direct comparison between No IS and IS groups. After 2 weeks, IS NICHE had viable, insulin-producing islets (Fig. 6m), whereas No IS NICHE only had limited presence of scattered cells (Fig. 6n),

suggesting islets had been destroyed. Mass cytometry (CyTOF) analysis of cell reservoir tissue showed IS NICHE had approximately 4-fold fewer infiltrating leukocytes than No IS NICHE (Fig. 6o). Moreover, of the infiltrating leukocytes, IS NICHE had proportionally ~7-fold less B cells and ~4-fold more NK cells than No IS NICHE; while CD4, CD8, and Treg populations did not differ between groups (Fig. 6p, Supplementary Fig. 7b, c). These results correlated with circulating cell populations (Supplementary Fig. 7d, e). Interestingly, the difference in NK populations was also present in the periphery pre-transplantation, suggesting an individual variation rather than a graft-dependent response (Supplementary Fig. 7f). Circulating CTLA4Ig levels peaked at 1.6 μg/mL on day 3 and decreased thereafter (Fig. 6q). In contrast, ATG and αCD154 had similar and steady plasma levels throughout the study, averaging 0.88 ± 0.18 μg/mL. Moreover, IS was enriched at the transplant microenvironment and tended to be higher, yet only statistically significant for αCD154, in the cell reservoir than the fibrotic

capsule (Fig. 6q). Taken together this data confirmed the NICHE is biocompatible, well tolerated, and effectively vascularized in nonhuman primates. Additionally, localized IS delivery with NICHE prolonged allogenic islet survival and modulated the local immune response, while significantly limiting systemic drug exposure.

## Discussion

In the work presented here, we performed efficacy testing of the NICHE: an implantable cell encapsulation platform integrating direct vascularization and localized immunosuppressant delivery for transplantation of allogeneic islets to treat T1D. The NICHE was designed to meet critical criteria for long-term islet engraftment and therapeutic effect, namely: (i) implant biocompatibility, (ii) islet accessibility to oxygen, nutrients, and metabolic products to maintain function, (iii) immune system evasion to prevent rejection, and (iv) reduced footprint and invasiveness to improve translatability and patient acceptability.

Biocompatibility and biointegration of NICHE are paramount for islet engraftment and function. We explored use of resin and PA for NICHE fabrication as these are commonly used in implantable medical devices[29]. Although both materials were biocompatible and minimally reactive (Fig. 1), resin NICHE integration was inconsistent; with only 25% of devices achieving full integration as opposed to 100% of PA NICHE. In general, the more mechanically similar a material is to adjacent body tissues, the better it is integrated[20]. The elastic moduli of both resin and PA were 6 orders of magnitude greater than that of the SubQ tissue[30]. The similarity in stiffness between resin and PA relative to the SubQ tissue suggests this parameter had little effect on material integration. Rather, PA was more hydrophobic than resin, which may increase macrophage adhesion and FBR[31,32]. Indeed, we observed a slight increase in macrophage content within the fibrotic capsule of PA compared to resin devices (Supplementary Fig. 2). The FBR to materials implanted subQ has been harnessed to create pre-vascularized sites for islet transplantation[4,33]. We posit that, in a similar fashion, NICHE biointegration harnessed the FBR to drive vascularization into the cell reservoir, which was further potentiated by the tissue remodeling properties of MSCs[19,34]. Furthermore, resin devices developed thin, yet dense and almost avascular fibrotic capsules, while PA fibrotic layers were about 2-fold thicker yet tended to be lax and vascularized. This was in line with previous reports showing a material roughness <1 μm, such as that of resin, promoted fibroblast spreading and dense fibrotic layers, whereas a roughness of 1–4 μm, observed with PA, had the opposite effect[35]. Thus, the density and composition, rather than the thickness, of the fibrotic layer may be better indicators of potential implant integration in the subQ space. Future studies on the specific mechanisms affecting implant biointegration will shed light on optimal materials for islet macroencapsulation. Ultimately, we confirmed PA is a suitable material for NICHE fabrication, integration, and vascularization.

Islet accessibility to oxygen, nutrients, and exchange of metabolic products was achieved through direct vascularization of the NICHE cell reservoir. In their native state, pancreatic islets are intimately connected to vasculature through intra-islet capillaries and receive ~20% of the pancreas blood supply albeit only comprising ~1% of the organ mass[12]. Thus, we aimed to create a microenvironment that would recapitulate this vascular support by leveraging the proangiogenic, wound healing, and tissue remodeling properties of MSCs during implantation and transplantation[36,37]. Of note, device implantation with MSCs leads to denser and more mature vascular networks than implantation with growth factor depots[19]. Indeed, the vascularized environment engineered within NICHE was conducive to long-term engraftment of functional islets, achieving diabetes reversal in rats for over 150 days. Furthermore, while hypovascularization of islet grafts in direct vascularization strategies is commonplace[38], NICHE had higher intra-islet vascular density than naïve pancreases. Interestingly, however, transplanted rats had a slight delay in glucose kinetics during intraperitoneal glucose tolerance testing, displaying peak BG levels at

30 min rather than at 15 min as typically observed in healthy animals. This is commonly observed with extrahepatic islet transplantation and is likely attributable to loss of direct insulin drainage into the portal circulation; thus, delaying insulin transport from the islets to the liver[39]. Notwithstanding, this delay did not seem to have a significant clinical impact on overall BG control or animal wellbeing.

Transplantation of allogeneic cells and tissues requires immune system evasion to prevent rejection. Clinical islet transplantation employs potent systemic immunosuppression to abrogate the immune response at the expense of life-threatening adverse effects including toxicity, opportunistic infection, and neoplasm development[2,3]. Here, we proved that localized immunosuppressant delivery with NICHE was as effective as the clinical standard of systemic delivery for allogeneic islet rejection prophylaxis in rats. Notably, NICHE delivery enriched immunosuppressants at the transplant microenvironment and limited systemic exposure by up to 36-fold, significantly reducing potential toxicity. Moreover, NICHE delivery did not induce measurable systemic immunosuppression, as evidenced by preserved populations of circulating lymphocytes and Tregs. This would have significant clinical impact in patient quality of life; not only by reducing the risk of opportunistic infection, but also by enabling boosting of the immune system against preventable diseases via vaccination, which lacks efficacy in immunocompromised patients[40,41]. Various local immunomodulation strategies for islet transplantation have been explored recently, including protein immobilization on biomaterial surface, chemical modification of the material, use of drug-releasing biomaterials, and co-transplantation with immunomodulatory cell types[42,43]. However, although promising, biomaterial and cell-based strategies have the remaining challenges of being limited in duration, difficult to replenish, and often rely on supplementation with systemic immunosuppression to effectively abrogate the allogeneic response[44]. To overcome these challenges, the NICHE was specifically designed for long-acting release and minimally invasive transcutaneous refilling via integrated silicone ports. This does not only allow for drug replenishment, but also for straightforward tailoring of the immunosuppressive cocktail throughout the graft lifespan as needed.

Local immunosuppression requires a shift in paradigm from standard systemic immunosuppression. In this context, advances in bio- and immunoengineering have influenced tailored approaches to tackle immune rejection specifically at the graft site, setting a new standard in the field of transplantation[45]. However, local immunosuppressant delivery as presented in this study remains largely unexplored. Thus, we decided to use a strategy that is systemically effective to allow for direct comparison between local and systemic administration. Since its introduction in the early 2000's, the Edmonton protocol, consisting of induction with a lymphocyte-depleting antibody followed by maintenance with sirolimus and tacrolimus, has been consistently used for clinical islet transplantation[46,47]. However, sirolimus and tacrolimus-related islet toxicity make them unsuitable for use in a localized setting[48–50]. Thus, our approach necessitated an immunosuppressive regimen that effectively abrogated immune rejection without causing islet toxicity. Alternative immunosuppression protocols using a combination of lymphocyte-depleting induction and costimulatory blockade maintenance have prevented islet allograft rejection in non-human primate models[51–54]. Of note, this immunosuppressive approach is the standard of care for kidney transplant rejection prophylaxis, underscoring clinical relevance and translatability. Therefore, in this study, we used ALS and CTLA4Ig for efficacy testing in rats. Here, both local and systemic immunosuppression effectively prevented allogeneic islet rejection and downregulated the memory immune response in rats. However, significantly lower systemic drug levels as well as preservation of circulating immune cell populations with NICHE delivery suggested mechanistic differences between local and systemic administration. We posit that localized delivery with NICHE-enriched immunosuppressants at the

transplant microenvironment, where CTLA4Ig acted on surveilling APCs, while ALS neutralized infiltrating lymphocytes. Concomitantly, as Tregs were preserved, these likely played a synergistic immuno-modulatory role for allograft protection[55]. Cross talk between MSCs and Tregs mutually enhances the immunosuppressive action of these cell types, further contributing to localized immunosuppression[56]. In contrast, lymphocyte depletion with systemic ALS essentially elimi-nated the rejecting cell pool in IP rats; while subsequent maintenance with high dose CTLA4Ig likely overwhelmed co-stimulation signaling and halted T-cell activation, preventing allograft rejection even in the context of reduced Tregs[57]. Of note, the plasma CTLA4Ig concentra-tions of IP rats were comparable to those reported in human patients after systemic dosing[58]. Therefore, as local delivery significantly lim-ited systemic exposure, it is unlikely allograft survival in NICHE rats was merely due to immunosuppressant permeation into systemic circulation.

In macaques, expecting a more stringent immunological barrier than in rats, αCD154 was added to the immunosuppressive cocktail. We theorized that tandem blockade of both main co-stimulation pathways would counteract the stringent, fully MHC-mismatched immunologi-cal barrier[59]. Local IS partially abrogated the immune response, as viable, insulin-producing islets were only observed in animals receiving IS and these grafts had 4-fold fewer infiltrating leukocytes than No IS controls. However, the proportion of regulatory, memory, and effector T cells were similar between IS and No IS animals both at the graft site and in the periphery (Supplementary Fig. 7), suggesting stalling rather than modulation of the immune response. The efficacy of combination therapy with systemic ATG and αCD154/CD40 pathway blockade has been documented[53,60,61], albeit the mechanism of action suggests heavy reliance on blockade at secondary lymphoid organs[62]. Based on our data (Figs. 4 and 6), local immunosuppression should focus on strin-gent lymphocyte depletion to hold off infiltrating T and B lympho-cytes, as well as polarization of the cell reservoir towards an immunosuppressive microenvironment (i.e., increasing Treg and M2 cells)[63,64]. This should be achieved with agents that act on infiltrating cells in situ, rather than through immunomodulation at secondary lymphoid organs. Importantly, drug enrichment at the transplant microenvironment via NICHE opens an avenue for localized use of potent immunosuppressive agents whose systemic use is limited by adverse effects, including those with the potential to guard off against autoimmune, antibody-based rejection. Ultimately, the results from this study unveiled an exciting path for exploration of local immuno-modulation strategies for islet allograft survival.

A limitation to our study is that not all rats became euglycemic, attributable to insufficient islet engraftment. In this study, islets were transplanted within a collagen matrix that provided temporary extra-cellular support to promote viability (Supplementary Fig. 8)[65]. Moving forward, the transplant matrix could be enhanced with growth factors or other extracellular matrix components such as fibrin to maximize islet engraftment[66,67]. It is noteworthy, however, that although not all rats became euglycemic, a significant benefit in non-fasting BG was observed for all animals transplanted in the NICHE and IP groups. This benefit in BG may translate into reduced exogenous insulin require-ments and glucose lability, which improve patient quality of life[68]. On another note, two NICHE animals rejected their grafts unexpectedly, which we attributed to device failure. Examination at endpoint showed these animals tended to have lower drug levels in cell reservoir tissue than others with viable grafts, suggesting insufficient drug release and concomitant inadequate immunosuppression (Supplementary Fig. 9). This could have resulted from faulty membrane porosity allowing rapid drug leakage and depletion. Moving forward, stringent quality control during device manufacturing in a clinical translation setting will help overcome these setbacks. Further, we showed localized immunosuppression prolonged islet survival in an MHC-mismatched non-human primate model. Future studies should extend onto

optimization of the local immunosuppressive cocktail and, subse-quently, assessment of long-term engraftment and diabetes reversal efficacy in a non-human primate, T1D model. Lastly, efficacy experi-ments were performed using only male rats as generalizability of islet transplantation strategies between sexes is not a known limitation in the field. Notwithstanding, recent studies report on sex-based differ-ences in islet biology whereby islets from females contain more insulin and respond faster to glucose stimulation ex vivo than islets from males[69]. Albeit notable, whether these differences affect islet trans-plantation efficacy in vivo is still under debate. As evidence continues to grow, future NICHE efficacy and characterization studies within the translation pipeline shall explore sex-based differences.

In the clinical context of islet allotransplantation, concomitant autoimmunity imposes an additional barrier for long-term islet survival. Ideally, islet allotransplantation studies as the one presented herein could benefit from an autoimmune diabetic model able to replicate the human natural history of disease, including autoimmunity recurrence after transplantation. However, models mimicking autoimmune T1D, including the spontaneous diabetic Wistar rat (BB rat), Komeda Diabetes-Prone (KDP) rat, and non-obese diabetic (NOD) mice, are far from ideal. The BB rat has heterogenous diabetes onset and progression of the disease, which compromises reproducibility[70]. Moreover, BB rats are lymphopenic and may accept allografts without need for an immu-nosuppressive regimen, directly contraposing the foundation of our study[71]. The autoimmune KDP rat model is unreliable as it develops insulitis with inconsistent severity, at an unpredictable time, and only in 70%[72] of animals. NOD mice are a well-established autoimmune model for T1D research[73]. However, the underlying mechanism for auto-immunity differs from that in humans and pre-clinical studies have shown poor translatability into the clinic[74]. Moreover, their use would impose severe technical restraints for long-term assessment of the NICHE (i.e. sampling frequency). Therefore, in our work, we used a streptozotocin-induced diabetic rat model, which is the preclinical standard for islet allotransplantation studies[75], including those with focus on immunosuppression[76–78]. Special interest should be given to the improvement of autoimmune T1D models as this will, in turn, benefit translatability assessment and optimization of new cell transplantation strategies such as the NICHE.

The NICHE was designed to reduce invasiveness and patient burden. Indeed, we demonstrated that all NICHE-related procedures could be conducted in an ambulatory setting in rats and nonhuman primates. The NICHE left a minimal imprint on skin after implantation, and fabrication in PA elicited a lower response than other materials used regularly in the clinic. NICHE deployment followed a carefully planned stepwise process to permit prevascularization, immunomo-dulation, and ultimately, islet engraftment. Our deployment approach aligns with current clinical islet transplantation protocols, where islet revascularization and multitargeted immunosuppression are required to achieve successful engraftment for T1D management. Moreover, SubQ implantation allows ease of access for repeated and straight-forward refilling via minimally invasive injections. In this study, we refilled the NICHE drug reservoirs every 28 days, a frequency com-parable to other long-acting drug formulations clinically available for applications such as migraine prevention or HIV treatment[79–81]. Nevertheless, reducing procedure periodicity by prolonging immu-nosuppressant release would reduce patient burden. This could be achieved by reducing the exchange surface area (e.g., membrane porosity) while increasing the amount of drug loaded[17]. Specifically, membranes with tightly controlled nanopore sizes as small as 2 nm have been used for sustained drug delivery[82–85]; while recent devel-opment of transcutaneous refilling of solid therapeutics has the potential to improve loading efficiency by 1000-fold[18]. The versatility in NICHE design allows straightforward incorporation of these mod-ifications. The resulting effects on drug biodistribution can be con-veniently estimated using our recently developed physiologically-

based pharmacokinetic model[86]. Finally, NICHE was resected en bloc, with intact fibrotic capsule and islets retained within the cell reservoir, underscoring feasible and safe retrieval.

In sum, the NICHE integrates direct vascularization and local immunosuppression into a single, implantable, replenishable device for allogeneic islet transplantation and long-term type 1 diabetes management. The results from this study demonstrate NICHE efficacy in rats and pave a path towards continued translational development of the approach. Overall, the NICHE is a promising solution with the potential to transform the field of islet transplantation for safe and prolonged treatment of type 1 diabetes.

## Methods
### NICHE fabrication
NICHE main structure was designed using a solid modeling software (SolidWorks 3D CAD v2021, Dassault Systèmes). The structure comprises a U-shaped drug reservoir (DR) that surrounds a central cell reservoir (CR). NICHE was fabricated by selective laser sintering (SLS) 3D printing (Sculpteo) using biocompatible polyamide (PA 2200, Electro Optical Systems) and by stereolithography (SL) 3D printing (Form 3b, Formlabs) using biocompatible resin (BioMed Clear, Formlabs). Two polyethersulfone (PES) nanoporous membranes (Sterlitech) were affixed between DR and CR, allowing for passive elution of immunosuppressants in the CR. The top and bottom surfaces of the CR were constructed by layering a set of 2 nylon meshes (Elko Filtering) per side on the main structure. The outer mesh had $100\,\mu m \times 100\,\mu m$ openings while the inner had $300\,\mu m \times 300\,\mu m$ openings. Ports for loading and refilling of CR and DR were made from implantable grade silicone adhesive (MED3-4213, Nusil) which was also employed in the assembling of membranes and meshes. NICHE components were autoclaved prior to assembly under a laminar flow hood using sterile technique. Fully manufactured NICHEs were sterilized with ethylene oxide gas at the Houston Methodist Research Institute Current Good Manufacturing Practice core. PA NICHEs were employed in the efficacy study (25 mm × 14.6 mm × 5 mm, CR: -500 μL), biocompatibility and vascularity assessment in rats and in vivo testing in NHP (30.4 mm × 15.4 mm × 3.8 mm, CR: -380 μL). Resin NICHEs (30 mm × 14.6 mm × 3 mm, CR: -300 μL) were employed in biocompatibility and vascularity assessment in rats. NICHEs implanted in rats and NHPs had 100 nm and 30 nm nanoporous membranes, respectively.

### Immunosuppressants
The immunosuppressants used for rat studies were CTLA4Ig (Abatacept, Bristol-Myers Squibb) and rabbit anti-lymphocyte serum (ALS; Accurate chemical); and for NHP studies anti-thymocyte globulin (ATG; Sanofi), CTLA4Ig (Belatacept, Bristol-Myers Squibb), and anti-CD154 [5C8H1] (NIH Nonhuman Primate Reagent Resource, RRID: AB_2716324).

### Scanning electron microscopy (SEM) imaging
Nylon meshes and membranes were sputtered with 7 nm iridium and imaged using Nova NanoSEM 230. Imaging was performed under high vacuum setting using a 5 kV electron beam at the Houston Methodist Research Institute Scanning Electron Microscopy and Atomic Force Microscopy Core.

### Roughness assessment
PA and Resin NICHE roughness was measured via atomic force microscopy (AFM). AFM contact mode in air scans were performed using the Biocatalyst Atomic Force Microscope (Bruker) and the MultiMode 8 Atomic Force Microscope (Bruker) for PA and Resin, respectively, at the Houston Methodist Research Institute Scanning Electron Microscopy and Atomic Force Microscopy Core. The average surface roughness was calculated on 9 $100\,\mu m \times 100\,\mu m$ fields of view per material via NanoScope Analysis Software (Bruker).

### Contact angle assessment
The water contact angle of PA and Resin NICHE samples (n = 3/material) was measured using the Attension Theta system (Biolin Scientific) for imaging and the One Attension software (Biolin Scientific) for analysis.

### Three-point flexural testing
PA and resin NICHE (n = 3/material) elastic moduli were assessed with a Univert (CellScale) mechanical testing machine using a 10 N load cell at a speed of 3 mm/min.

### In vitro drug release
CTLA4Ig was labeled with Alexa-Fluor 647 NHS ester (Invitrogen) according to the manufacturer's instructions. Conjugated CTLA4Ig (abatacept) was mixed with unlabeled CTLA4Ig at a 1:9 ratio and diluted to 55 mg/mL using USP water for injection (RM Bio). IgG-FITC conjugate (Sigma Aldrich) was diluted to 2 mg/mL using USP water for injection. Syringes with 25G needles were used to inject CTLA4Ig and IgG solutions in NICHE drug reservoirs integrated with 100 nm PES nanoporous membranes (n = 4/molecule). Loaded NICHEs were individually submerged in glass scintillation vials containing 22 mL of phosphate-buffered saline (PBS) and incubated at 37 °C under constant agitation. The sink solution was collected and fully replenished every third day. Samples of the sink solution were analyzed in duplicates using a plate reader to measure fluorescence (Synergy H4, Biotek).

### Animal models
Eight-week-old male Lewis (Charles River Strain Code 004, MHC Haplotype RT1$^l$) and Fischer 344 (F344; Charles River Strain Code 002, MHC Haplotype RT1$^{lv}$) rats were used as donors and recipients, respectively. All animals were maintained and used in conformity with guidelines established by the American Association for Laboratory Animal Science. Rats were kept in the Comparative Medicine Program facility at Houston Methodist Research Institute in environmentally controlled rooms with a standard 12 h dark/light cycle. Animals were housed in pairs, fed Teklad global 18% protein diet (Envigo), and provided water ad libitum. Welfare monitoring was performed daily. Marked lethargy, hypothermia, severe dehydration, ataxia, hunched posture, labored breathing, tiptoe or slow ponderous gait, infection at sight of implant, wound dehiscence, loss of body weight ≥20% compared to baseline, and BCS < 2 were established as humane endpoints. Upon study endpoints, euthanasia was performed via $CO_2$ asphyxiation. All procedures were approved by the Houston Methodist Institutional Animal Care and Use Committee (IACUC, #IS00005894) in accordance with the National Institute of Health Guide for the Care and Use of Laboratory Animals and the Animal Welfare Act.

For NHP studies, a 5-year-old male Mauritian cynomolgus monkey (*Macaca fascicularis*) purchased from the Mannheimer Foundation, Inc. (Homestead, FL) through the University of Miami Division of Veterinary Resources (DVR) was used as donor. The donor NHP was kept at the University of Miami animal facility. Female Vietnamese cynomolgus macaques (World Wide Primates Inc.), were used for biocompatibility studies (n = 2, ages 11, 12 years) and as recipients for islet transplant studies (n = 4, ages 11, 12, 13, 14 years). Recipient NHPs were kept at the AAALAC-I accredited Michale E. Keeling Center for Comparative Medicine and Research, The University of Texas MD Anderson Cancer Center (UTMDACC). Donor-recipient pairs were ABO compatible and mismatched as possible for MHC I and II alleles, identified via microsatellite analysis or deep sequencing[87]. All animals were pair-housed in environmentally controlled rooms and fed a standard laboratory diet with constant access to clean, fresh water. Welfare monitoring was performed daily. Humane endpoints included infection, ulceration or wound dehiscence at implant site, signs of systemic inflammatory response syndrome (i.e. hypothermia, fever,

tachycardia, tachypnea, leukocytosis, or leukopenia), anorexia, loss of body weight ≥20% compared to baseline, and BCS < 2. Upon study endpoints euthanasia was performed using humane practices (IV pentobarbital) recommended by the American Veterinary Medical Association Guidelines of Euthanasia. All experiments were carried out according to the provisions of the Animal Welfare Act, PHS Animal Welfare Policy, and the principles of the NIH Guide for the Care and Use of Laboratory Animals. All procedures were approved by the IACUC at UTMDACC (#000017-49RN00) and Mannheimer Foundation (#2021-01).

### Biocompatibility assessment

In rats, medical grade titanium (Ti) and resin discs (1 mm thick × 8 mm diameter) and resin and PA NICHE were implanted subQ for 6 weeks. In NHPs, PA NICHE were implanted subQ for 4–6 weeks. Ti (20 mm × 13 mm × 4.5 mm) devices implanted subQ for 16 weeks were used as reference[88]. Retrieved implants were fixed and fibrotic capsules processed for histology. The thickness of the fibrotic capsule was measured on Masson's Trichrome-stained tissue sections with ImageJ software. At least 10 randomly selected areas were measured for each implant. Technical replicates were averaged, and biological replicates were pooled for comparison between groups. For implant reactivity scoring, H&E-stained tissue sections were scored by a blinded, board-certified pathologist using an established scoring system (Supplementary Table 1)[88,89].

### Generation, implantation, and retrieval of NICHE with MSCs

For rat studies, MSCs isolated from bone marrow of F344 rats were obtained from Cyagen (Cat No. RAFMX-01001) at P2 and expanded in vitro using StemXVivo Mesenchymal Stem Cell Expansion Media (R&D Systems). For NHP studies, autologous MSCs were isolated from whole blood using published methods[19,90]. Briefly, whole blood was collected from each animal in sodium heparin tubes (BD). The mononuclear cell fraction was isolated using Ficoll and cultured in DMEM (Sigma) supplemented with 0.1 mM nonessential amino acids, 100-U/ml penicillin and streptomycin, 1 ng/ml basic fibroblast growth factor (FGF), 10% fetal calf serum (FCS), and 0.2 mM L-glutamine (Gibco). Adherent cells positive for markers CD105, CD90, CD29 and negative for markers CD14, CD34, CD45 were defined as MSCs. On implantation day, MSCs were suspended in a pluronic F-127 (Sigma) hydrogel (20% PF-127 in DMEM) and injected into the cell reservoir of NICHE ($5 \times 10^5$ per NICHE). MSCs cell permanence within NICHE was assessed (Supplementary Fig. 1).

For implantation in rats, F344 rats were anesthetized using 1–2% isoflurane and appropriate anesthesia depth was confirmed by the absence of pedal withdrawal reflex. Next, a 2 cm incision was made to create a subQ pocket, and NICHEs implanted in the dorsum. The wound was closed using clips, and rats recovered under heat supplementation until motor skills were regained. For implantation in NHPs, animals were pre-anesthetized with ketamine and anesthesia was maintained with 1–2% isoflurane. NICHE was inserted into a subQ pocket in the dorsum through a 2 cm incision. The subQ tissue and skin were closed with an absorbable monofilament suture and animals recovered.

For retrieval, a 2 cm incision was made in anesthetized animals and NICHE resected en bloc with fibrotic capsule via blunt dissection. The subQ tissue and skin were closed with an absorbable monofilament suture and animals recovered.

### Vascularity assessment

Resin and PA MSC-NICHE were implanted subQ in F344 rats for 6 weeks ($n = 4$/material). Tissue sections of explanted NICHE were stained with *B. simplicifolia* lectin (L2140, Sigma, 1:100) to label blood vessels. For NHPs, 2 PA MSC-NICHE were implanted subQ in cynomolgus macaques ($n = 2$ NICHE/NHP). After 4 and 6 weeks, 1 NICHE was

removed from each NHP, processed for histology, and tissue sections stained with CD31 (ab28364, Abcam, 1:50). Stained sections were visualized using EVOS M5000 imaging system coupled with a 40× objective (Life Technologies). Five and ten fields were randomly imaged per slide in rats and NHPs, respectively, and blood vessels were quantified by a blinded scientist. Blood vessel density was calculated as number of blood vessels per $mm^2$. For rats, technical replicates were averaged, and biological replicates pooled for comparison between groups. For NHPs, all technical replicates per timepoint were pooled. Five fields of view per NHP, per timepoint were randomly selected and vessel area was calculated using Eq. 1.:

$$\text{Vessel area}(\%) = \frac{\text{Area occupied by vessels}}{\text{Total section area}} \times 100 \qquad (1)$$

### Rat islet isolation

Lewis rats were overdosed with isoflurane immediately prior to pancreas harvesting. Pancreas were perfused through the pancreatic duct with 9 mL CIzyme RI collagenase (Vitacyte) and 0.2 µg/mL DNAse (dornase alfa; Genentech) in Hanks balanced salt solution (HBSS; Gibco) with 10 mM HEPES (Gibco). Digestion was performed via static incubation in a water bath at 37 °C for 19 min and 40 s. Next, enzymatic activity was quenched by adding 20 mL of 20% fetal bovine serum (FBS; Gibco) in HBSS and mechanical dissociation performed via vigorous shaking for 8 s. Tissue digest was washed 3 times with ice cold HBSS and strained through a 500 µm wire mesh. The tissue pellet was re-suspended in 15 mL 1.096 g/cm³ OptiPrep (Sigma) and layered with 10 mL 1.068 g/cm³ and 1.037 g/cm³ OptiPrep and centrifuged at $480 \times g$ for 20 min with no break. Islets were collected at the 1.096 g/cm³ and 1.068 g/cm³ interphase, washed with HBSS 3 times and cultured in RPMI-1640 media supplemented with 10% FBS, 1% penicillin/streptomycin, 20 mM HEPES, 5.5 mM glucose and 1 mM sodium pyruvate (all from Gibco) in a cell culture incubator set at 25 °C and 5% $CO_2$.

### NHP islet isolation

For donor pancreatectomy, a midline abdominal incision from the xyphoid process to the pubis was made. The animal was exsanguinated, utilizing a 14 G catheter in the aorta, and the pancreas was dissected from the spleen, portal vein, and duodenum, with ligation of the pancreatic ducts. The pancreas was placed in Euro Collins Solution (Corning) and taken to the lab. Islet cell isolation was performed using modifications of the automated method for human islet isolation[91]. The tissue was digested with Mammalian Tissue Free Liberase (Liberase MTF C/T, Roche), and discontinuous Euroficoll gradients (densities: 1.132; 1.108; 1.096; 1.037) were used for purification of islets from the pancreatic digest. The tissue was bottom-loaded with stock Ficoll and centrifuged in a COBE 2991 blood cell processor (Terumo).

### In vitro glucose stimulated insulin release (GSIR)

Krebs buffered solution (KBS) was prepared with 115 mM NaCl, 24 mM NaHCO₃, 5 mM KCl, 1 mM MgCl₂, 0.1% bovine serum albumin (BSA), and 2.5 mM CaCl₂ (all from Fisher Scientific) and 25 mM HEPES (Gibco). Low (LG) and high (HG) glucose KBS was supplemented with 2.8 mM or 16.7 mM glucose (Gibco), respectively. Islets were transferred to 3 µm cell culture inserts in a 24-well plate and equilibrated with LG-KBS 1 h at 37 °C. Next, inserts were transferred to 15 mL falcon tubes and centrifuged at 100 rcf for 15 s. Inserts with islets were transferred to a clean 24-well plate and incubated with LG-KBS for 1 h at 37 °C followed by incubation with HG-KBS for 1 h at 37 °C. The KBS was collected after each incubation step and analyzed for insulin content via ELISA (Alpco). The stimulation index was calculated as the HG-to-LG insulin secretion ratio.

### In vitro drug toxicity assay

After incubation with CTLA4Ig, ALS or culture media (Vehicle) for 5 days, islet function was assessed via GSIR as described here.

### Tube formation assay

HUVECs (C-12200, PromoCell) were incubated with low serum growth supplement (LSGS) supplemented media for 5 days. Next, cells were harvested and resuspended in non-supplemented media in the presence of ATG (0.05 mg/mL and 0.1 mg/mL), CTLA4Ig (0.5 mg/mL and 1 mg/mL), or a combination of CTLA4Ig 1 mg/mL and ATG (0.05 mg/mL or 0.1 mg/mL). Supplemented media and Suramin (30 μM, Sigma) were used as positive inducer and inhibitor of tube formation, respectively. Non-supplemented media was used as control (Veh). Matrigel was distributed in a 24-well plate and incubated for 30 min at 37 °C to permit gelling. HUVECs were prepared at a density of $4.1 \times 10^5$ cells/mL and 300 μL of cell suspension was added into each well. After overnight incubation at 37 °C, cells were stained with Calcein AM (Invitrogen) and tube formation assessed using a Keyence BZ-X800 Fluorescence Microscope (Keyence). Four pictures per well were taken using the 4X objective and GFP filter. Images were analyzed using the Angiogenesis Analyzer plugin in ImageJ software.

### Real-time quantitative polymerase chain reaction (qPCR)

Following overnight treatment with ATG and CTLA4Ig, HUVEC cells were washed twice with 1× PBS. Total ribonucleic acid (RNA) was extracted with Trizol (Life Technologies) following the manufacturer's protocol. Total RNA was measured via NanoDrop One (Thermo Scientific). Real-time (RT) qPCR was performed using 100 ng of total RNA, TaqMan Fast Virus 1-Step Master Mix (Applied Biosystems), and TaqMan gene expression assay (Supplementary table 2). All RT-qPCR assays were performed with biological and experimental duplicates using StepOnePlus Real Time system (Applied Biosystems). Gene expression was calculated by relative quantitation expressed as fold change (2-ΔΔCt), in relation to untreated control samples (Veh).

### Efficacy study in rats

F344 were implanted with MSC-NICHE subQ. After 2 weeks of implantation, rats were rendered diabetic via intraperitoneal (i.p.) streptozotocin (STZ; Sigma) injection of 65 mg/kg divided in 2 doses of 40 mg/kg and 25 mg/kg 5 days apart. BG was measured daily until diabetes was confirmed with 3 consecutive BG readings >300 mg/dL. After 6 weeks of vascularization, rats were randomized into 3 immunosuppressive groups (day 0): NICHE, IP, and No IS. NICHE rats ($n = 12$) received NICHE drug reservoir loading with 9 mg CTLA4Ig (30 mg/mL) and 0.6 mg ALS (2 mg/mL) for local delivery. IP rats ($n = 8$) received an induction regimen consisting of ALS (10 mg, i.p. day 0) and CTLA4Ig (20 mg/kg, i.p., days 0, 3, 6, 9, 12, 15) followed by maintenance with weekly CTLA4Ig (20 mg/kg, i.p.). No IS rats ($n = 4$) served as controls and did not receive immunosuppression. A cohort of healthy rats ($n = 5$) was used as reference. On day 3, all rats were co-transplanted with 3000 islet equivalents (IEQ) and $4.5 \times 10^6$ syngeneic MSCs into the NICHE cell reservoir (Supplementary Fig. 3). On day 31, all rats received a second islet-MSCs transplant to boost therapeutic effect. NICHE rats drug reservoirs were refilled on days 28, 56, and 157. IP rats received a second ALS injection on day 28 in preparation for re-transplantation. Body weight and BG were measured weekly, and euglycemia was determined as BG < 200 mg/dL. Islet function was assessed via intraperitoneal glucose tolerance test (IPGTT) on days 112 and 151, and fasting C-peptide quantification on day 112 via ELISA (Crystal Chem). Overall glycemic control was assessed via HbA1c quantification (Crystal Chem) at euthanasia for non-euglycemic rats and upon NICHE removal for euglycemic rats. NICHE ($n = 4$), IP ($n = 3$), and No IS ($n = 4$) rats that did not achieve euglycemia were euthanized on day 63 and NICHE, fibrotic capsule, skin, and peripheral tissues (spleen, liver, kidney, lymph node, and blood) collected for analysis. Rats that reverted to diabetic state (BG > 200 mg/dL) were euthanized and same tissues collected. On day 115, NICHEs were explanted from 2 of 5 euglycemic NICHE rats and 5 of 5 euglycemic IP rats in a survival surgery. On day 193, NICHEs were explanted from the remaining 3 NICHE rats. Upon removal, NICHEs were processed for histology and drug quantification. Rats were euthanized 2 days after NICHE removal and peripheral tissues collected for analysis. Circulating lymphocytes were measured on days 0, 3, 7, 14, 21, 28, 31, 56, 84, and 115, and Tregs on day 84 via flow cytometry.

### NICHE assessment in NHPs

Cynomolgus macaques ($n = 4$) were implanted subQ with MSC-NICHE. After 6 weeks of vascularization, NICHEs of 3 NHPs were loaded with an immunosuppressant cocktail (ISC) consisting of 9 mg (30 mg/kg) CTLA4Ig, 3 mg (10 mg/kg) ATG, and 3 mg (10 mg/kg) anti-CD154 (day 0). One NHP did not receive ISC loading and served as a control. On day 3, 2000 allogeneic IEQ and $1 \times 10^7$ autologous MSCs were co-transplanted in NICHE of all NHPs. Blood was collected on days 0, 3, 10, and 17 for ISC quantification and on days 0, 3, and 17 for peripheral blood immunophenotyping. On day 17, NICHEs from all NHPs were explanted in a survival surgery and processed for histology, ISC quantification, and mass cytometry (CyTOF) analysis.

### Islet transplantation

Islet doses were aliquoted into 15 mL falcon tubes and allowed to sediment (Supplementary Fig. 3). An MSC-collagen hydrogel was prepared by re-suspending MSC in 4 mg/mL neutralized, thermo-responsive collagen hydrogel (Advanced Biomatrix). Rat and NHP MSCs were procured as described in subsection *Generation and implantation of NICHE with MSCs*. Islet supernatant was carefully aspirated out without disturbing the islet pellet. Islets were re-suspended with MSC-collagen hydrogel using a wide tip and immediately loaded into a 1 mL syringe equipped with a 22 G needle. The islet-MSC-hydrogel was immediately transcutaneously injected into NICHE cell reservoir through the silicone port. To prevent leakage, the needle was held in place for 3 min to allow for hydrogel polymerization.

### NICHE drug reservoir loading

Immunosuppressant solution was loaded in a 1 mL syringe coupled with a 25G needle. The syringe was advanced through the skin and a lateral silicone port into the drug reservoir. A second 25G needle was advanced in the same fashion through the contralateral silicone port to vent and prevent pressure buildup. Once both needles were in place, immunosuppressant solution was slowly pushed into the drug reservoir.

### Immunosuppressant quantification

In rats, whole blood was collected in lithium heparin microtainers (BD) and plasma isolated via centrifugation. In NHPs, whole blood was collected in SST vacutainers (BD) and serum isolated via centrifugation. Tissues were homogenized using T-PER buffer (Thermo Scientific) supplemented with Protease Inhibitor Tablets (Thermo Scientific). CTLA4Ig was quantified using CTLA-4 Soluble Human ELISA Kit (Invitrogen); ALS was quantified with Easy-titer Rabbit IgG (Thermo Scientific); ATG was quantified with Rabbit Fc kit (Perkin Elmer); and anti-CD154 was quantified with Human IgG Kappa MAb kit (Perkin Elmer).

### BG monitoring and IPGTT

BG levels were assessed via tail prick with a commercial veterinary glucometer (AlphaTrack, Zoetis). For IPGTT, rats were fasted for 16 h and injected i.p. with 3 g/kg of 50% dextrose solution (Northeast Medical Products). BG readings were obtained at baseline and at 15, 30, 60, 90, and 120 min post injection.

## Histology and immunohistochemistry

Upon harvesting, tissues were fixed in 10% formalin for 5 days. Fixed tissues were dehydrated in standard ethanol and xylene washes followed by paraffin embedding. Tissues were cut in 5 μm sections and stained with hematoxylin-eosin (H&E) and Masson's Trichrome using standard technique at the Research Pathology Core of Houston Methodist Research Institute. For immunohistochemistry, sections were stained with *Bandeiraea simplicifolia* lectin, CD31, or insulin using standard protocols[17,19]. For lectin staining, following deparaffinization and rehydration, sections were blocked with 5% normal goat serum (NGS) in 0.1% bovine serum albumin/tris buffered saline (BSA/TBS) for 1 h at room temperature, followed by overnight incubation with biotinylated lectin (L2140, Sigma, 1:100) in 0.1% BSA/TBS at 4 °C and subsequent incubation with streptavidin alkaline-phosphatase (434322, Invitrogen, 1:100) for 30 min at RT. Sections were developed with Warp Red Chromogen system (WR806, Biocare Medical, California, USA) and counterstained with hematoxylin following the manufacturer's instructions. Alternatively, following deparaffinization, rehydration, heat-induced antigen retrieval, and blocking in 5% NGS, sections were incubated with CD31 (ab28364, Abcam, 1:50) or insulin (C27C9, Cell Signaling, 1:100) antibodies for 1 h at room temperature, followed by incubation with HRP-conjugated goat anti-rabbit secondary antibody and developed using 3,3'-diaminobenzidine (DAB) system (Pierce, 34002). Sections were visualized using EVOS M5000 imaging system (Life Technologies).

## Effect of local immunosuppressant release on angiogenesis in vivo

F344 rats ($n = 4$) were implanted with 2 NICHE (1 in each flank), loaded with $5 \times 10^5$ BM-MSCs and vascularized for 6 weeks. For each rat, one NICHE drug reservoir was loaded with 9 mg CTLA4Ig (30 mg/mL) and 0.6 mg ALS (2 mg/mL) for local Immunosuppressant (IS) release NICHE. The contralateral NICHE drug reservoir was loaded with saline and served as a control. Two weeks later, NICHE were harvested and processed for histology and VEGF quantification. For functional blood vessel staining, following pressure-cooker-based epitope retrieval, sections were stained with primary antibodies CD31 (NB100-2284, Novus Biologicals, 1:200), VE Cadherin (36-1900, Invitrogen, 1:25), and eNOS (ab300071, Abcam, 1:50). Anti-rabbit Alexa Flour 555 (A-21428, Invitrogen, 1:200) was used as secondary antibody. Imaging was performed using a Nikon Eclipse TE300 Fluorescent Microscope in the FL2 channel (585 ± 20 nm). Fluorescence intensity was measured by 2 independent observers using the NIS-Elements Basic Research software. VEGF was quantified from tissue lysate using Bio-Plex Pro Rat Cytokine VEGF Set (#171L1026M) magnetic bead-based assay (Bio-Rad Laboratories) on the Bio-plex platform (Bio-Rad), according to manufacturer's instructions.

## Islet revascularization assessment

Pancreas or cell reservoir tissue sections were stained with *B. simplicifolia* lectin and visualized using EVOS M5000 imaging system coupled with a 40× objective (Life Technologies). The islet area within a field of view was isolated and the percent islet area covered by blood vessels quantified. Biological replicates were pooled for comparison between groups.

## Imaging mass cytometry (IMC) analysis

IMC analysis was performed at the ImmunoMonitoring Core at the Houston Methodist Research Institute. Antibodies were conjugated with metals following the Fluidigm protocol as previously described[92] (Supplementary table 3). For staining, NICHE sections were baked at 60 °C overnight followed by deparaffinization in xylene and rehydration in alcohol gradients (absolute ethanol, absolute ethanol:deionized water 90:10, 80:20, 70:30, 50:50, 0:100) for 10 min each. Epitope retrieval was done in a water bath at 95 °C in

Tris-Tween 20 buffer pH 9 for 20 min. Sections were blocked with 3% BSA in tris-buffered saline, followed by staining overnight at 4 °C with markers in Supplementary table 1 and nuclear staining with Cell-ID Intercalator (Fluidigm). Slides were air-dried and ablated with Hyperion (Fluidigm) for data acquisition. Data were segmented by ilastik and CellProfiler. Histology topography cytometry analysis toolbox (HistoCAT) and R scripts were used to quantify cell number and generate tSNE plots[93]. Analysis was performed on 2 ROI/sample and $n = 3$/group for a total of $n = 6$ ROI/group.

## CyTOF analysis

To prepare single cell suspensions, NICHE cell reservoir tissue was cut into 2 mm × 2 mm pieces and digested using collagenase/hyaluronidase in DMEM (Stem Cell) at 37 °C for 35 min with orbital shaking at 100 rpm. Digestion was quenched with 2% FBS in PBS and digest filtered through a 40 μm strainer. Cells were pelleted and RBC lysed with ACK lysing buffer (Quality Biological). Cells were washed twice and re-suspended with 2% FBS in PBS. Single-cell suspension was stained with metal-tag viability dye for 5 min and washed with cell staining buffer (Fluidigm), followed by sequential staining of surface and intracellular markers detailed in Supplementary table 4. Next, cells were stained with Cell ID Intercalator Ir (Fluidigm) at 4 °C overnight. The next day, cells were washed and prepared for acquisition with Helios (Fluidigm). The data was analyzed by Cytofbank. Briefly, data was normalized, gated out beads and dead cells, and gated on singlet and CD45⁺ cell population, which was selected to perform a tSNE analysis. The tSNE plots were gated according to the gating strategy described in Supplementary table 5. The ratio of cell populations within CD45⁺ cells was quantified by Cytobank and plotted with Prism (Graphpad).

## Flow cytometry analysis

**Rat model.** For memory T cell analysis at endpoint, spleens were disaggregated into single cell suspensions via mechanical dissociation in PBS with 5 mM EDTA and filtered through a 40 μm cell strainer. Red blood cells were removed via incubation with ACK lysis buffer (Quality Biological). Cells were washed and, following incubation with FC blocker (αCD32, BD Biosciences), $1 \times 10^6$ cells were stained with CD45, CD3, CD4, CD8a, CD44, CD62L, and viability dye. Cells were washed with 2% FBS in PBS with 5 mM EDTA and fixed with IC fixation buffer (Invitrogen). For blood analysis, 100 μL of whole blood anticoagulated with EDTA was stained with same markers as the spleen followed by incubation with 1-step Fix/Lyse Solution (eBioscience). Cells were transferred to 5 mL FACS tubes with 40 μm cell strainer (Corning) and washed twice with 2% FBS in PBS. For circulating lymphocyte analysis, 100 μL whole blood was stained as above with CD45 and CD3. For circulating Treg analysis, 100 μL of whole blood was stained as above with CD45, CD3, CD4, CD25, and viability dye, followed by fixation with eBioscience fixation/permeabilization buffer (Invitrogen), permeabilization with 1× permeabilization buffer (Invitrogen), and intracellular staining with Foxp3. Cells were washed twice and re-suspended in 2% FBS in PBS with 5 mM EDTA. Data was acquired on LSRII flow cytometer equipped with FACSDiva v9 software (BD Biosciences) and analyzed using FlowJo v10 software (FlowJo, LCC). Analysis was performed after exclusion of debris, doublets and nonviable cells. Effector memory T cell (Tem) population was defined as CD3⁺CD4⁺CD44^hiCD62L^lo or CD3⁺CD8⁺CD44^hiCD62L^lo. Central memory T cell population (Tcm) was defined as CD3⁺CD4⁺CD44^hiCD62L^hi or CD3⁺CD8⁺CD44^hiCD62L^hi. Circulating lymphocytes were defined as CD45⁺CD3⁺. Treg population was defined as CD3⁺CD4⁺CD25⁺Foxp3⁺. Antibodies used are detailed in Supplementary table 6. Gating strategy exemplified in Supplementary Fig. 10.

**NHP model.** Immunophenotyping of whole blood were undertaken via immunofluorescent staining and flow cytometric analysis with

antibodies specified in Supplementary Table 7 [43]. All flow cytometry samples included a Live/Dead Fixable Near-IR Dead Cell Stain (Molecular Probes) for viability assessment and were run on a Becton Dickinson LSR II cytometer and analyzed using Kaluza software (version 1.5a, Beckman Coulter). Cell populations were defined as follows: Effector memory T cell (Tem) as $CD3^+CD4^+CD95^+CD28^-$ or $CD3^+CD8^+CD95^+CD28^-$; Central memory T cell population (Tcm) as $CD3^+CD4^+CD95^+CD28^+$ or $CD3^+CD4^+CD95^+CD28^+$; Natural killer cells as $CD3^-CD16^+CD8^+$; B cells as $CD20^+$; Tregs as $CD3^+CD4^+CD25^+Foxp3^+$. Gating strategy exemplified in Supplementary Fig. 11.

### Statistical analysis

Results are expressed as mean ± standard deviation. Statistical analyses were performed using Prism software (GraphPad Software Inc.). Normality testing was performed and Student's $t$-test, one-way analysis of variance (ANOVA) and post-hoc analyses were used to determine statistical significance of differences among groups. Specific analysis method, number of replicates, and $p$ values are specified in each figure legend.

### Reporting summary

Further information on research design is available in the Nature Portfolio Reporting Summary linked to this article.

## Data availability

All data generated or analyzed during this study are included in this published article (and its supplementary information files). Source data are provided with this paper.

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

## Acknowledgements

The authors are grateful to Dr. Marco Farina for insightful discussion and information. The authors thank Dr. Jianhua (James) Gu from the electron microscopy core of Houston Methodist Research Institute; Dr. Andreana L. Rivera, Yuelan Ren, and Sandra Steptoe from the research pathology core of Houston Methodist Research Institute; Drs. Yitian Xu, Licheng Zhang, and Shu-Hsia Chen from the ImmunoMonitoring Core of Houston Methodist Research Institute; Dr. David L. Haviland and Nicole Vaughn from the flow cytometry core of Houston Methodist Research Institute; Alex Rabassa and Waldo L. Diaz from the Diabetes Research Institute at the University of Miami for pancreas procurement and islet isolation; and Luke Segura, Elizabeth Lindemann, Dana Salazar, Bharti Nehete, and Dr. Greg Wilkerson from the Michale E. Keeling Center for Comparative Medicine and Research at UTMDACC for support in NHP studies. Figure 3a was partially created with biorender.org. The authors also thank graphic designer Virginia Facciotto (virginia.facciotto@gmail.com) for the preparation of schematics and illustrations. Funding support from Juvenile Diabetes Research Foundation 1-INO-2018-595-A-N (A.G.), Vivian L. Smith Foundation (A.G.), Houston Methodist Research Institute (A.G.), Diabetes Research Institute (N.K.), and in part by NIH NIDDK R01DK132104 (A.G., J.N.).

## Author contributions

J.P-M. designed and conceptualized the study, designed and performed experiments, collected, analyzed, and interpreted data, prepared figures, and wrote the manuscript. J.N.C-C. designed and performed experiments, collected and analyzed data, prepared figures, and contributed to manuscript writing. S.C. designed and performed experiments, collected and analyzed data, and provided conceptual advice. C.Y.X.C. provided conceptual advice, interpreted data, and contributed to manuscript writing. N.H., H-C.L., F.P.P-F., G.M., B.A., J.A.N., L.B.A., and M.A.W. performed experiments, collected, and analyzed data. K.A.S. and S.K. performed experiments and collected data. P.N.N. performed experiments and provided conceptual advice. D.M.B. performed experiments, interpreted data, provided essential material and conceptual advice. X.C.L., C.R., and A.O.G. provided essential material and conceptual advice. J.E.N and N.S.K. interpreted data, acquired funding, provided essential material and conceptual advice. A.G. conceived, conceptualized, and supervised the study, acquired funding, and revised the manuscript. All authors have read and approved the manuscript.

## Competing interests

J.P-M., C.Y.X.C., S.C., and A.G. are inventors of intellectual properties licensed by NanoGland, LLC. The remaining authors declare no competing interests.

## Additional information

[1]Department of Nanomedicine, Houston Methodist Research Institute, Houston, TX, USA. [2]School of Medicine and Health Sciences, Tecnologico de Monterrey, Monterrey, NL, Mexico. [3]University of the Chinese Academy of Sciences (UCAS), Shijingshan, Beijing, China. [4]Center for Tissue Engineering, Houston Methodist Research Institute, Houston, TX, USA. [5]Department of Comparative Medicine, Michael E. Keeling Center for Comparative Medicine and Research, MD Anderson Cancer Center, Bastrop, TX, USA. [6]The University of Texas Graduate School of Biomedical Sciences at Houston, Houston, TX, USA. [7]Diabetes Research Institute, University of Miami, Miami, FL, USA. [8]Department of Surgery, Miller School of Medicine, University of Miami, Miami, FL, USA. [9]Department of Surgery, Houston Methodist Hospital, Houston, TX, USA. [10]Immunobiology and Transplant Science Center, Houston Methodist Hospital, Houston, TX, USA. [11]Department of Microbiology and Immunology, Miller School of Medicine, University of Miami, Miami, FL, USA. [12]Department of Biomedical Engineering, University of Miami, Miami, FL, USA. [13]Department of Biochemistry and Molecular Biology, University of Miami, Miami, FL, USA. [14]Department of Radiation Oncology, Houston Methodist Hospital, Houston, TX, USA. ✉e-mail: agrattoni@houstonmethodist.org

