## [Peer Review File · Nature Communications]

Implantable niche with local immunosuppression for islet allotransplantation achieves type 1 diabetes reversal in rats.REVIEWER COMMENTS

Reviewer #2 (Remarks to the Author):

This manuscript describes the application of a revascularized and locally immunosuppressive device for islet transplantation. Biochemical and cellular approaches have been applied for immunomodulation in both small and large animals. This study introduced a platform that integrates direct vascularization and local immunosuppression into a single, implantable, replenishable device for allogeneic islet transplantation and long-term type 1 diabetes.

The manuscript is written concisely and easy to read. However, there are several questions as to whether the data supports the conclusion. Some of the details may be provided to make the paper a better step for the subsequent clinical translations.

comments:

1. The route of immunosuppressants release is from drug reservoir to cell reservoir. Since the results (Fig.1 m-o) show that immunosuppressants at high concentrations are toxic to islet function, why not let them be directly released outside the device but not through the cell reservoir? It may help protect islets by achieving a wilder immune-suppressing in the cell reservoir.

2. In figure 1 n, islets treated with ALS at 1 mg/ml showed a lower functionality, although it's not a significant difference. In all of fig 1 m-n, the vehicle groups should be treated with the same condition, why do the basal stimulatory indexes have big differences? The basal stimulatory index in figure m is around 2 while in figure o it became 8? These immunosuppressant toxicity tests may be reperformed.

3. What are the optimal working concentrations for these immunosuppressants? Whether or not the fluctuations in concentration result in graft dysfunction and/or host target immunocyte tolerance. Please explain and discuss.

4. Inflammation can promote angiogenesis depending on T cell activation and Th1 polarization. Does the applied immunosuppressant inhibit angiogenesis? Please check the effects of CTLA4Ig and ALS on vascular endothelial cell signaling and angiogenesis.

5. How can authors compare the vessel area with only 1/ 4 sample of the resin NICHE fully integrated while all PA devices were integrated? In lines 111-112, the authors claimed 'Vascular density quantification of resin (n = 2) and PA (n = 4)'. Is the resin device sample filled by clotted blood able to be statistically analyzed? Only 3 dots were shown in Fig 1k PA group, please check the 'n'.

6. The retrieved devices show different thicknesses of the walls. The resin ones' are thinner, while PA ones' are thicker. Is this difference influence biocompatibility?

7. The mechanical characteristics of resin and PA devices may be compared.

8. In line 248 authors regard a-SMA+ area as blood vessels, however, as shown in figure 3 a and b, most a-SMA+ cells surround islet graft and do not form tubular structures. They may consist of vascular

pericyte and myofibroblast.

9. The cell revivors were wrapped by a set of 2 nylon meshes with 100 μm x 100 μm and 300 μm x 300 μm openings, respectively. Although these pore sizes may facilitate host tissue and vessel growth in, I just wonder if there are leakages of islet graft and MSCs. Especially when MSCs were suspended in a pluronic F-127 gel (line 626), which will change from a solid into liquid at physiological temperature.

10. In figure 4 b, immunosuppressants were refilled on day 157. Although there was a significant plasma concentration increase of ALS and CTLA4IG, the concentrations cannot restore as the first two doses induced. Please explain. The release efficiency may decrease as time goes by? Does this long-term release deficiency compromise its potential clinical applications?

11. The analysis of NICHE biocompatibility, islet engraftment, and local immunomodulation in NHP is successful. The implantations resulted in great graft islet functionality and abundant vasculatures. These implantations support NICHE effectively protecting allotransplanted islets in primates. However, the NICHE devices designed for rats and NHPs are similar sizes. Is it enough to potentially restore blood glucose of a diabetic NHP? Please discuss the device's capacity to manage glycemia of large animals as well as human beings.

12. In lines 561-562, 'NICHEs implanted in rats and NHPs had 100 nm and 30 561 nm nanoporous membranes, respectively.' Why nanoporous membranes with different pore sizes were used in rats and NHPs?

13. Scale bar was missing in Supplementary Fig. 7.

Reviewer #3 (Remarks to the Author):

Paez-Mayorga et al. describe a 3D-printed polyamide-based encapsulation device (NICHE) that provides local immunosuppression via drug reservoirs to protect islets loaded into the cell reservoir from rejection while facilitating integration and neovascularization. NICHE's advantage over other encapsulation devices is its ability to deliver immunosuppressive agents locally, which enables it to be used in a broad variety of scientific applications.

Although the biocompatibility, vascularization, and sustained local immunosuppressive drug delivery function of the NICHE device were previously reported in 2020 Biomaterials, the current study regarding clinical translation is significant and well worth reporting in a high-profile journal, if the listed limitations are convincingly addressed. The manuscript contains some limitations, include a small sample size, the absence of autoimmunity, and the absence of islet function measurements in the NHP model. Despite these limitations, the findings are exciting, with substantial clinical implications.

Below are some suggested comments to improve the paper.

Major comments:

1) For a rodent study, the number of animals per NICHE and IP groups is relatively small. If the purpose is to demonstrate efficacy of local immunosuppressive drug delivery to prevent allogeneic islet graft rejection, biocompatibility, and vascularization of NICHE, then more animals are needed. The number of rats engrafted, vascularized, and reversed diabetes over a 150-day period is between 3-4 rats per group. Note: The figure legends do not match the statement made in the manuscript regarding the number of rats per groups (see minor comments).

2) The ongoing autoimmunity is a major barrier for allograft acceptance. The efficacy of NICHE was tested in chemically diabetic rat recipients, the lack of autoimmunity weakens the translational aspects of the NICHE for islet transplantation.

3) Although the NHP studies to test the biocompatibility and engraftment for the translation purposes are promising, but it does not address islet engraftment and function over time; since the NHPs were not rendered diabetic and given that NICHE was explanted after 14 days of islet load leaves NICHE's suitability for long-term islet engraftment unanswered in NHPs.

4) NICHE implant acceptability and biocompatibility in IS NHP recipients are impressive, but the percentage of NK and CD8 positive cells infiltrates in comparison to recipients without immunosuppressive drugs raises concerns about drug delivery efficiency; perhaps this could be addressed by improving loading efficiency.

5) Quantification of islet mass in rats and IS NICHE NHPs should be included to determine long-term engraftment efficiency and islet load requirement.

6) Fig. 1e) What is the implant reactivity after 6 weeks? Did the authors assess the implant reactivity at the study endpoint? Even though the difference between Titanium and PA at 6 weeks is not significant, PA had higher reactivity. It would be highly beneficial if the data could be made available and supplemented.

Minor comments:

1) Was the foreign body reaction to NICHE assessed at the study endpoint? it would be very valuable if the data is available and added.

1) For translational purposes, the islet loading capacity of NICHE to normalize BGL in diabetic recipient should be discussed.

2) ALS drug release kinetics from NICHE were similar to CTLA4Ig?

3) The time of STZ injection in Fig2a does not match in the Line 174-175_”..... At 2 weeks post-implantation, rats were rendered diabetic via streptozotocin injection” Two weeks after implantation is Day -28, figure 2a shows Day -10 for STZ injection.

4) Fig 2d legend does not match the manuscript line 215-218. Fig2d legend states “.....reservoir receiving local (NICHE; n = 8 to day 91, n = 5 to day 116, n = 3 to day 195)”. Whereas in line 215-218 states “....Two rats in the NICHE group rejected their grafts prematurely on day 84 (leaving 6 rats) due to immunosuppressant release failure and were removed from study. On day 115, NICHE was explanted from remaining IP rats and 2 of 5 NICHE rats, while the remaining 3 NICHE rats..... Remaining number of rats should be 4 not 3?.....

REVIEWER COMMENTS

We thank the reviewers for their interest and insightful comments on our work. During the revision period, we have invested significant time and resources to perform additional experimental work in vitro and in vivo to address each one of the comments. Specifically, new experiments were performed to assess immunosuppressant toxicity on islets, immunosuppressant effect on angiogenesis via tube formation capacity, angiogenic gene and protein expression, and vascular maturity and function, subchronic implant reactivity, mechanical characterization of the material, and cell retention analysis. We also expanded the discussion on our prototyping of NICHE for pre-clinical and clinical testing and its implications moving the platform forward. Addressing the observations and recommendations from the reviewers has significantly strengthened our manuscript. Here, we present a point-by-point response to the reviewer's comments.

Reviewer #2 (Remarks to the Author):

This manuscript describes the application of a revascularized and locally immunosuppressive device for islet transplantation. Biochemical and cellular approaches have been applied for immunomodulation in both small and large animals. This study introduced a platform that integrates direct vascularization and local immunosuppression into a single, implantable, replenishable device for allogeneic islet transplantation and long-term type 1 diabetes. The manuscript is written concisely and easy to read. However, there are several questions as to whether the data supports the conclusion. Some of the details may be provided to make the paper a better step for the subsequent clinical translations.

1. The route of immunosuppressants release is from drug reservoir to cell reservoir. Since the results (Fig.1 m-o) show that immunosuppressants at high concentrations are toxic to islet function, why not let them be directly released outside the device but not through the cell reservoir? It may help protect islets by achieving a wilder immune-suppressing in the cell reservoir.

We thank the reviewer for this interesting observation. The key concept behind localized immunosuppression with NICHE is to thwart activation of the immune responses at the transplant microenvironment where antigen presenting cells pick up antigen and where effector cells infiltrate to destroy the graft. Thus, by localizing immunosuppressants within the cell reservoir, immune rejection is prevented without systemic adverse effects. Conversely, immunosuppressant delivery outside of the device would decrease distribution within the graft and likely necessitate higher doses to prevent rejection. This would result in increased systemic drug exposure, defeating the purpose of local immunosuppression.

2. In figure 1 n, islets treated with ALS at 1 mg/ml showed a lower functionality, although it's not a significant difference. In all of fig 1 m-n, the vehicle groups should be treated with the same condition, why do the basal stimulatory indexes have big differences? The basal stimulatory index in figure m is around 2 while in figure o it became 8? These immunosuppressant toxicity tests may be reperformed.

The toxicity assays presented in the original version of the manuscript were performed independently of each other using various batches of isolated islets. We attribute the differences in stimulatory indexes to intrinsic variability between islet batches. For this reason, statistical significance was calculated relative to the vehicle control for each drug combination and batch presented in Fig. 1m-n. Acknowledging the reviewer's concern, we performed new toxicity assays using a single batch of islets and relevant immunosuppressants doses. The new data set recapitulated the results obtained previously. Fig. 1 m-n has been revised to show the updated data.

3. What are the optimal working concentrations for these immunosuppressants? Whether or not

the fluctuations in concentration result in graft dysfunction and/or host target immunocyte tolerance. Please explain and discuss.

We thank the reviewer for this insightful query. In the work presented herein, we showed that ALS and CTLA4Ig were effective in preventing rejection and were not toxic to islets at the concentrations used in vivo. However, the exact optimal doses of ALS and CTLA4Ig in a local setting are still to be determined. Localized immunosuppression for allograft protection as presented in our work has not been previously reported, limiting the breath of literature accessible for comparison. Across our various studies, we have consistently observed that while drug concentration may slowly decrease over time, grafts remain viable and functional. Thus, we hypothesize that the drug doses may be further scaled down while maintaining effective immunosuppression. Optimization of the local immunosuppressant doses for allogeneic islet rejection prophylaxis is part of our next steps in the investigation of the NICHE platform.

4. Inflammation can promote angiogenesis depending on T cell activation and Th1 polarization. Does the applied immunosuppressant inhibit angiogenesis? Please check the effects of CTLA4Ig and ALS on vascular endothelial cell signaling and angiogenesis.

We thank the reviewer for this interesting observation. The effect of the CTLA4 pathway on endothelial cell signaling has been explored in the context of rheumatic arthritis, cancer, and atherosclerosis. CTLA4Ig, which recapitulates native CTLA4 signaling, decreased VEGFR and ICAM expression, suggesting a negative effect in angiogenic potential¹. However, anti-CTLA4 mAb, which blocks native CTLA4 signaling, increased endothelial inflammation, and enhanced atherosclerotic plaque formation². Moreover, anti-CTLA4Ig mAb seems to have a synergistic anti-angiogenic effect with anti-VEGF mAb to decrease tumor burden³. Taken together, these studies suggest that both CTLA4 pathway engagement and blockade may reduce angiogenesis. Anti-thymocyte globulin (ATG; or rat analogue ALS) increased angiogenesis after cardiac infarction in rats, attributed to pro-angiogenic cytokine release from dying leukocytes⁴. However, a direct effect of ATG on endothelial cell signaling has not been reported.

To address this comment in the context of our platform, we performed extensive new experimental work in vitro and in vivo to assess the effect of CTLA4Ig and ATG/ALS on angiogenesis. For in vitro studies, high and low doses of each drug were selected based on their in vitro release profiles and tissue concentrations measured at study endpoint (Fig. 1l and 5d). We found that CTLA4Ig, ATG and CTLA4Ig+ATG at any dose did not affect tube formation capacity of human endothelial cells in terms of tube length or number of segments (Fig. 2a-c). We further characterized the effect of these immunosuppressants on angiogenesis via gene expression analysis. High- or low-dose CTLA4Ig, ATG and CTLA4Ig+ATG did not affect the expression of angiogenic genes *cdh5*, *nos3*, and *vegf* compared to vehicle controls (Fig. 2d). High-dose, but not low-dose, ATG induced a 4-fold increase in *vcam* expression, a gene associated with endothelial inflammation⁵. However, high dose ATG was nearly 50 times higher than the concentration found in NICHE cell reservoirs, suggesting in vivo ALS release maintained local concentration below the endothelial inflammatory threshold. We further characterized the effect of these immunosuppressants on angiogenesis in vivo. F344 rats were implanted subQ with 1 NICHE on each flank and allowed to vascularize for 6 weeks as per our standard protocol. Next, one of the two NICHE was transQ filled with CTLA4Ig+ALS for local release, while the contralateral NICHE was filled with saline and served as a control. Two weeks later, NICHE were harvested and assessed for angiogenic signaling and vascularization. Quantification of VEGF levels served as a surrogate of angiogenic signaling while immunohistochemical labeling with CD31, VE-Cadherin and eNOS informed on vessel maturity and function. In line with in vitro gene expression analysis, VEGF levels were similar in the microenvironment of control and locally immunosuppressed NICHE (Fig. 2e). Additionally, immunohistochemical analysis showed CD31, VE-Cadherin, and eNOS expression was comparable between control and locally immunosuppressed NICHE ($p > 0.5$ across groups), indicating that NICHE vasculature was mature and functional independently of local CTLA4Ig-ALS release (Fig. 2f-g). Taken together, this data indicated that CTLA4Ig and ATG/ALS release did not affect angiogenic signaling,

vessel maturity or function. We have incorporated this new data as an additional manuscript figure (Fig. 2) as well as in results subsection “Effect of local immunosuppressant release on angiogenesis”.

5. How can authors compare the vessel area with only 1/ 4 sample of the resin NICHE fully integrated while all PA devices were integrated? In lines 111-112, the authors claimed ‘Vascular density quantification of resin (n = 2) and PA (n = 4)’. Is the resin device sample filled by clotted blood able to be statistically analyzed? Only 3 dots were shown in Fig 1k PA group, please check the ‘n’.

We appreciate the reviewer’s observation. Of the resin NICHE only 1 / 4 was fully integrated, 1 / 4 had some tissue ingrowth but was mainly filled with clotted blood, and 2 / 4 were not integrated at all. The vascular density quantification of the resin NICHE was made on the fully integrated and clotted samples. Upon further reflection, we recognize the clotted sample is not suitable for comparison and has been eliminated. Fig. 1k PA was missing replicate, we have included the replicate and updated the figure as well as the results section accordingly.

6. The retrieved devices show different thicknesses of the walls. The resin ones’ are thinner, while PA ones’ are thicker. Is this difference influence biocompatibility? The mechanical characteristics of resin and PA devices may be compared.

We appreciate the reviewer’s observation. One of the main factors that triggers the foreign body reaction (FBR) to an implanted material is the material stiffness in relation to body tissues⁶. In general, the more mechanically similar a material is to body tissues, the lesser of a FBR will be elicited. To put this in the context of the NICHE, we assessed the mechanical characteristics of PA and resin devices via three-point flexural testing. The elastic modulus of resin was 3.07 ± 0.14 GPa and of PA was 1.79 ± 0.17 GPa ($p = 0.0012$) (See figure below). For reference, the subcutaneous (subQ) tissue has a stiffness of 4.85 kPa⁷. Overall, the elastic modulus for both resin and PA were 6 orders of magnitude greater than that of the SubQ tissue. Therefore, we posit that the slight difference in wall thickness between the resin and PA NICHE had little effect on the FBR elicited towards resin and PA devices. We believe that the differences in contact angle and material roughness, previously described, might have had a more prominent effect on material integration within the SubQ tissue. It is noteworthy that even though resin and PA NICHE experienced different FBR and subQ integration, both materials were biocompatible, with tissue reactivity scores comparable to those of medical-grade Ti implant controls. We have included the resin and PA elastic modulus in results subsection “NICHE fabrication and characterization” as well as comment in this regard in the discussion.

Elastic modulus of resin and PA assessed via three-point flexural testing. The elastic modulus of subQ tissue was obtained from Iatridis et al., *Connect. Tissue Res.*, 2003.

7. In line 248 authors regard a-SMA+ area as blood vessels, however, as shown in figure 3 a and b, most a-SMA+ cells surround islet graft and do not form tubular structures. They may consist of vascular pericyte and myofibroblast.

We thank the reviewer for this clarification. We have revised the text to describe α SMA⁺ cells as “vascular cells”.

9. The cell revivors were wrapped by a set of 2 nylon meshes with 100 μ m x 100 μ m and 300 μ m x 300 μ m openings, respectively. Although these pore sizes may facilitate host tissue and vessel growth in, I just wonder if there are leakages of islet graft and MSCs. Especially when MSCs were suspended in a pluronic F-127 gel (line 626), which will change from a solid into liquid at physiological temperature.

F127, which we use as a vehicle for MSCs, gels at a physiological temperature of 37 °C, favoring cell permanence within the NICHE after implantation. To shed light onto the reviewer’s query, we performed new experiments assessed permanence of F127-loaded cells within NICHE in vitro. Briefly, we suspended DiI-labeled MSCs in 20% PF127 and loaded them into NICHE cell reservoirs following the same procedure to prepare devices for implantation described in our methods section. Loaded NICHE were placed in a cell culture incubator at 37 °C to allow PF127-MSCs to gel and incubated overnight to assess cell leakage from the devices. After incubation for 24 h, NICHE were transferred to a neighboring well and both wells were imaged with IVIS (Supplementary Fig. 1). Of note, the hydrogel was visibly still intact within the cell reservoir. Upon IVIS imaging, fluorescent signal was only detected in the NICHE-containing well, demonstrating that the cells remained confined within NICHE. This observation correlated with our previous work showing bioluminescent cell permanence within NICHE after 4 weeks of transplantation⁸. In terms of islets, these have an average diameter of 150 μ m which is superior to the inner mesh of 100 μ m, limiting the probability of leakage⁹. Moreover, at time of islet transplantation, NICHE have vascular tissue ingrowth into the cell reservoirs which serves as an additional barrier against islet leakage. Finally, in this study we showed that NICHE explanation reverted euglycemic rats to a diabetic state, further demonstrating that islets were contained within NICHE (Fig. 3d). Taken together, our multiple in vitro and in vivo assessments confirm that cells are contained within NICHE. The results from our recent in vitro testing are included in Supplementary Fig. 1 and referenced to in the results subsection “NICHE fabrication and characterization”.

10. In figure 4 b, immunosuppressants were refilled on day 157. Although there was a significant plasma concentration increase of ALS and CTLA4IG, the concentrations cannot restore as the first two doses induced. Please explain. The release efficiency may decrease as time goes by? Does this long-term release deficiency compromise its potential clinical applications?

We thank the reviewer for this comment. We believe that the difference in restoration of drug concentrations in plasma upon refilling of NICHE after long-term implantation may be due to tissue remodeling inside NICHE. We posit that as islets engraft and extracellular matrix is deposited, tissue density inside NICHE increases. This, in turn, may affect drug diffusivity within and outside the NICHE and into systemic circulation, resulting in lower plasma levels in favor of enhanced localization. We expect this tissue remodeling to plateau over time and eventually remain steady. As we move our work forward, we plan on performing longitudinal characterization of tissue remodeling inside NICHE and its effect on drug distribution. It is noteworthy that one of the key features of NICHE is refillability. We aim to leverage this feature for straightforward adjustment of immunosuppressant loading doses to maintain target release profiles within the remodeled microenvironment in pre-clinical and eventual clinical deployment. Moreover, we believe that the difference in restoration of drug levels in plasma over time may indicate enhanced drug localization, emphasizing the novelty of our work related to the efficacy of localized immunosuppression through NICHE for immunoprotection of islet allografts to revert diabetes.

11. The analysis of NICHE biocompatibility, islet engraftment, and local immunomodulation in NHP is successful. The implantations resulted in great graft islet functionality and abundant vasculatures. These implantations support NICHE effectively protecting allotransplanted islets in primates. However, the NICHE devices designed for rats and NHPs are similar sizes. Is it enough

to potentially restore blood glucose of a diabetic NHP? Please discuss the device's capacity to manage glycemia of large animals as well as human beings.

We appreciate the reviewer's observation. Due to the scope of the study, we decided to employ rat-sized devices in NHPs. The goal of our first-in-NHP study was to assess NICHE biocompatibility and the potential for localized immunosuppression to thwart the allogeneic response to transplanted islets. Employing a rat-sized device permitted implantation of multiple devices per NHP, maximizing the amount of data collected while minimizing the number of animals needed. The rat-sized device is estimated to hold up to 44,000 IEQ. For NHPs and humans, a curative islet dose oscillates between 5000 to 15000 IEQ/kg, which translates to roughly 25,000 – 75,000 IEQ for an average macaque and 750,000 IEQ for a human. Thus, an NHP device with approximate dimensions of 37 mm x 14.6 mm x 3 mm would be necessary to deliver a curative islet dose. We have also projected the dimensions of human NICHE to be 60 mm x 36 mm x 3 mm to hold about 350,000 IEQ. Based on this, we estimate that two NICHE would be sufficient for transplantation of a curative islet dose. To put this in the context of other macrodevices that have reached clinical trials, the Cell Pouch by Sernova is 30% larger than NICHE and the VC-02-20 from Viacyte was tested with implantation of up to 4 devices with a capacity of 250,000 IEQ each¹⁰.

12. In lines 561-562, 'NICHEs implanted in rats and NHPs had 100 nm and 30 561 nm nanoporous membranes, respectively.' Why nanoporous membranes with different pore sizes were used in rats and NHPs?

We thank the reviewer for the observation. Based on the analysis of our rat studies, we hypothesized that scaling down local immunosuppressant doses could be suitable (comment 3). In this context, we have performed a separate study to characterize the release and biodistribution of different sized molecules from NICHE using various membrane pore sizes. The results from this study are currently under revision for publication. Confidentially, we can share with the reviewers that as part of our findings, we observed that a membrane porosity of 30 nm prolonged the sustained release of full-size antibody molecules (See figure below). Thus, we explored this configuration in our NHP study.

In vitro IgG release from NICHE through a 30 nm membrane over 30 days. Release was calculated as a percent of the total cargo loaded.

13. Scale bar was missing in Supplementary Fig. 7.

We thank the reviewer for the observation; the scale bar has been added. This is now Supplementary Fig. 8.

Reviewer #3 (Remarks to the Author):

Paez-Mayorga et al. describe a 3D-printed polyamide-based encapsulation device (NICHE) that provides local immunosuppression via drug reservoirs to protect islets loaded into the cell reservoir from rejection while facilitating integration and neovascularization. NICHE's advantage over other encapsulation

devices is its ability to deliver immunosuppressive agents locally, which enables it to be used in a broad variety of scientific applications.

Although the biocompatibility, vascularization, and sustained local immunosuppressive drug delivery function of the NICHE device were previously reported in 2020 Biomaterials, the current study regarding clinical translation is significant and well worth reporting in a high-profile journal, if the listed limitations are convincingly addressed. The manuscript contains some limitations, include a small sample size, the absence of autoimmunity, and the absence of islet function measurements in the NHP model. Despite these limitations, the findings are exciting, with substantial clinical implications.

Below are some suggested comments to improve the paper.

We appreciate the reviewer's appreciation for our platform and their recognition of its substantial clinical implications. The reviewer's comments were quite insightful, and we paid close consideration into each one in our revised work. These queries were addressed via thorough literature review of state-of-the-art models of diabetes, in-depth analysis and interpretation of our results, and projections for clinical application for the NICHE. Furthermore, we performed new in vivo work focusing on subchronic NICHE reactivity. Finally, have included confidential data from a separate study, currently under revision for publication, to shed clarity onto some inquiries. We have addressed each comment below.

Major comments:

1) For a rodent study, the number of animals per NICHE and IP groups is relatively small. If the purpose is to demonstrate efficacy of local immunosuppressive drug delivery to prevent allogeneic islet graft rejection, biocompatibility, and vascularization of NICHE, then more animals are needed. The number of rats engrafted, vascularized, and reversed diabetes over a 150-day period is between 3-4 rats per group.

Note: The figure legends do not match the statement made in the manuscript regarding the number of rats per groups (see minor comments).

We thank the reviewer for this query. The main endpoints for our study were diabetes reversal via islet transplantation in NICHE and efficacy of local vs systemic IS in preventing long-term rejection. We achieved significantly improved glycemic profile in 100% of transplanted rats in the NICHE and IP groups, and diabetes reversal in 66.7% (8/12 animals) and 62.5% (6/8 animals) of NICHE and IP rats, respectively (Supplementary Fig. 1 and Fig. 3d). These rats were studied for at least 100 days, a time threshold commonly accepted in the field of transplantation as long-term or even indefinite graft function in rodents. To explore engraftment even further, as a minor endpoint, we maintained 3 NICHE rats (i.e. receiving local immunosuppression) close to 200 days. Albeit the number of animals for this secondary endpoint might have been limited, our overall work employed animal numbers comparable to other islet transplant studies recently published in Nature Communications¹¹⁻¹⁴. We believe that, just as the aforementioned studies, our work presented herein is substantiated and will contribute significantly to the field of islet transplantation.

2) The ongoing autoimmunity is a major barrier for allograft acceptance. The efficacy of NICHE was tested in chemically diabetic rat recipients, the lack of autoimmunity weakens the translational aspects of the NICHE for islet transplantation.

We agree in principle that the studying of our system in an autoimmune diabetic model able to replicate the human disease would be ideal. However, after a thorough search on the topic, we found no reliable or appropriate autoimmune diabetic models suitable to test the NICHE. In fact, existing models mimicking autoimmune type 1 diabetes, including the spontaneous diabetic Wistar rat (BB rat), Komeda Diabetes-Prone (KDP) rat, and non-obese diabetic (NOD) mice, have critical flaws and disadvantages.

The BB rat is a spontaneous autoimmune diabetic model with heterogenous diabetes onset and progression of the disease. Even though BB rats have insulinitis with infiltration of T cells, B cells, and macrophages, they are lymphopenic with extreme decrease in CD4+ T cells¹⁵. Due to this state of low immunological response, diabetic animals accept syngeneic or allogeneic islet grafts without any immunosuppressive regimen¹⁶. This defies the purpose of testing the NICHE in this model.

The autoimmune KDP rat model does not reliably develop diabetes. Further, insulinitis develops with an inconsistent severity (moderate to severe) and at an unpredictable time within 220 days of age and only in 70%¹⁷ of animals. These factors cannot be controlled, which renders the model not suitable for the reproducible assessment of our technology.

NOD mice are the most-established autoimmune model for T1D research¹⁸. However, it is not without notable limitations. Histopathology in NOD mice differs significantly from patients with T1D¹⁹, and therapies developed with NOD mice are not translatable into treating patients with T1D²⁰. Moreover, the NOD model would not only require us to move backward with respect to clinical translation relevance, but it would also be subject to significant experimental challenges and expensive to use. In this context, as opposed to focusing our translational efforts towards a non-human primate model, we would reverse into mice testing, which tends to be less relevant than rats or larger animals for T1D research²¹. To study the NICHE in mice, we would need to redesign and fabricate the NICHE to be substantially smaller and suitable for implantation, rendering this effort even more challenging. Moreover, we would need to retest tolerability of the NICHE under the mice skin, re-evaluate time and process of NICHE neovascularization, re-establish PK and PD of drug to determine the suitable dose of immunosuppressant drugs to be used, and reassess islet dose. In all of this, we would be severely limited by the volume and frequency of blood samples that can be collected in such a small animal, which is part of the reason why we focused the NICHE developments on rats and not mice. Overall, work with the NOD mouse model would necessitate a ~2-year effort, unavailable significant funding resources, and major practical limitations.

In summary, there are no reliable or appropriate autoimmune diabetic models for the NICHE. In our work we leverage the use of a well-established chemically (streptozotocin) induced diabetic rat model, which is a preclinical standard for islet allotransplantation studies and the most widespread used model in literature²², including studies such as ours that have a focus on immunosuppression²³⁻²⁵. We have included comment on the aforementioned points in the discussion of the revised manuscript.

3) Although the NHP studies to test the biocompatibility and engraftment for the translation purposes are promising, but it does not address islet engraftment and function over time; since the NHPs were not rendered diabetic and given that NICHE was explanted after 14 days of islet load leaves NICHE's suitability for long-term islet engraftment unanswered in NHPs. The main goal of the pilot NHP study was to demonstrate NICHE translatability potential by means of biocompatibility, vascularization, and islet transplantation testing. We agree with the reviewer that full-scale studies to assess NICHE efficacy for diabetes treatment in NHP are needed, and these are within our future plans. However, the complexity of efficacy testing in a diabetic NHP model falls outside the scope of the present study and would require significant resources currently unavailable to us.

4) NICHE implant acceptability and biocompatibility in IS NHP recipients are impressive, but the percentage of NK and CD8 positive cells infiltrates in comparison to recipients without immunosuppressive drugs raises concerns about drug delivery efficiency; perhaps this could be addressed by improving loading efficiency.

We agree with the reviewer that the immunosuppressive effect in NHPs was not as potent as that observed in the rodent model. We attribute this to a more complex immunological barrier present in NHPs. Therefore, we believe that moving towards translation, an optimal immunosuppressive cocktail is to be elucidated. This might require a combination of various immunosuppressive drug classes as well as

synergistic local and systemic IS. The elucidation of the aforementioned parameters is one of the primary goals of our endeavors as we move the NICHE forward. We have discussed this in our manuscript.

5) Quantification of islet mass in rats and IS NICHE NHPs should be included to determine long-term engraftment efficiency and islet load requirement.

We appreciate the reviewer's enquiry and recognize this is an important aspect to elucidate. As islets disperse throughout the cell reservoir, reliably quantification of the islet mass in NICHE requires homogenization of the entire cell reservoir tissue. Although this was something we considered, doing so would have prevented us from performing other key analyses for characterization of our platform including flow cytometry, histology, and imaging mass cytometry. Moreover, using the NICHE for multiple analyses allowed us to maximize scientific output while reducing the number of animals needed in accordance with ethical considerations on the 3R's in experimental research. This is, however, part of our future studies.

6) Fig. 1e) What is the implant reactivity after 6 weeks? Did the authors assess the implant reactivity at the study endpoint? Even though the difference between Titanium and PA at 6 weeks is not significant, PA had higher reactivity. It would be highly beneficial if the data could be made available and supplemented.

We thank the reviewer for this suggestion. We were unable to assess implant reactivity at study endpoint as the fibrotic capsule tissue was employed for drug distribution analyses (Fig. 5d). However, recognizing the relevance of the reviewer's suggestion to extend beyond 6 weeks, we conducted a new in vivo study to assess implant reactivity in rats for 12 weeks. The NICHE reactivity score at 12 weeks of implantation was 10.67 ± 4.37 compared to 10.25 ± 2.63 at 6 weeks ($p = 0.8799$). The stark similarity in scores suggests that host reactivity against NICHE was not exacerbated over time. This is in line with recent literature reviews by our group and others concluding that, after an initial acute reaction, reactivity towards an implant material eventually reaches a steady state^{6,26}. The timing to reach said steady state may vary depending on various device characteristics. A previous study by our group found stabilization of the foreign body reaction of vascularized polymeric devices after around 4 weeks of implantation in rats and non-human primates²⁷. Ultimately, this data shows that the NICHE is swiftly and steadily integrated into the host subQ tissue. We have included this new data in Supplementary Fig. 1b-c.

Minor comments:

1) Was the foreign body reaction to NICHE assessed at the study endpoint? it would be very valuable if the data is available and added.

We appreciate the reviewer's suggestion. Due to the limited amount of tissue available from the fibrotic capsule, it was entirely used for drug quantification analysis (Fig. 5d). Therefore, the FBR was not assessed at endpoint. However, newly performed analysis on NICHE implanted for up to 12 weeks found that the FBR did not change significantly over time (Supplementary Fig. 1b-c). Long-term NICHE biocompatibility will be assessed in future studies using chronic implantation models in large animals.

2) For translational purposes, the islet loading capacity of NICHE to normalize BGL in diabetic recipient should be discussed.

We thank the reviewer for the suggestion. We refer the reviewer to our reply to concern 11 by reviewer number 2. Briefly, the human NICHE prototype has dimensions 60 mm x 36 mm x 3 mm and could hold about 350,000 IEQ. Based on these characteristics, we estimate 1 to 2 NICHE would be sufficient to normalize BGL in a diabetic recipient.

3) ALS drug release kinetics from NICHE were similar to CTLA4Ig?

The ALS and CTLA4Ig release kinetics were assessed in vitro (Fig. 1i). From this data, we found that CTLA4Ig tended to have a slight burst release while ALS displayed a steadier release, attributable to the

difference in molecule size (98 kDa for CTLA4Ig vs. 150 kDa for ALS). As part of our characterization efforts on release kinetics, we have further investigated the release and biodistribution of CTLA4Ig and full antibody sized molecules from NICHE in vivo. The results from this independent study are currently under revision for publication. Confidentially, we share with the reviewers that the in vivo data showed both molecules had similar release kinetics and biodistribution (see figure below). This suggests that, when loaded at similar concentrations, the release and biodistribution of these drugs did not change significantly.

a cumulative drug release from NICHE in vivo was calculated based on the amount of drug remaining in the drug reservoir at time of explant. CTLA4Ig $R^2 = 0.7806$ and slope = 1.622; IgG $R^2 = 0.7828$ and slope = 1.449. Biodistribution of b CTLA4Ig and c IgG in tissues after 14 days of release.

4) The time of STZ injection in Fig2a does not match in the Line 174-175_”..... At 2 weeks post-implantation, rats were rendered diabetic via streptozotocin injection”.... Two weeks after implantation is Day -28, figure 2a shows Day -10 for STZ injection.

We thank the reviewer of the observation. We amended the figure to reflect the accurate timeline which is STZ injection two weeks after implantation; this is now Fig. 3a.

5) Fig 2d legend does not match the manuscript line 215-218. Fig2d legend states “.....reservoir receiving local (NICHE; n = 8 to day 91, n = 5 to day 116, n = 3 to day 195)”. Whereas in line 215-218 states “....Two rats in the NICHE group rejected their grafts prematurely on day 84 (leaving 6 rats) due to immunosuppressant release failure and were removed from study. On day 115, NICHE was explanted from remaining IP rats and 2 of 5 NICHE rats, while the remaining 3 NICHE rats..... Remaining number of rats should be 4 not 3?.....

We thank the reviewer of the observation. One rat from the NICHE group that had remained euglycemic was injured during cage swapping procedures within our animal facility and had to be euthanized on day 90. Thus, the animal distribution is as follows: on day 84 two rats were removed due to immunosuppressant release failure, leaving 6 rats; on day 90, 1 rat was euthanized due to injury, leaving 5 rats; on day 115, NICHE was explanted from 2 rats, leaving 3 rats. We apologize for the confusion resulting from omission of the rat injury. We have included detail on this matter to increase clarity of the timeline in the results subsection “NICHE efficacy testing for allogeneic islet transplantation and diabetes reversal”.

Regarding the figure legend, it states the number of data points that are potted for the graph independently of the glycemic status of the rats. For instance, even though 2 rats rejected their grafts on day 84, the plot also includes the hyperglycemic readings for these two rats on day 91. Similarly, even though 2/5 rats were explanted on day 115, we report their day 116 glycemic readings showing reversal

to diabetic state. The figure legend did not account for the rat that was removed due to injury. To correct for this oversight, we have amended the figure legend as follows: "NICHE; n = 8 to day 84, n = 7 to day 91, n = 5 to day 116, n = 3 to day 195". This is now Fig 3.

References

1. Cutolo, M., *et al.* CTLA4-Ig/CD86 interactions in cultured human endothelial cells: effects on VEGFR-2 and ICAM1 expression. *Clin Exp Rheumatol* **33**, 250-254 (2015).
2. Poels, K., *et al.* Antibody-Mediated Inhibition of CTLA4 Aggravates Atherosclerotic Plaque Inflammation and Progression in Hyperlipidemic Mice. *Cells* **9**(2020).
3. Wu, X., *et al.* Combined Anti-VEGF and Anti-CTLA-4 Therapy Elicits Humoral Immunity to Galectin-1 Which Is Associated with Favorable Clinical Outcomes. *Cancer Immunol Res* **5**, 446-454 (2017).
4. Lichtenauer, M., *et al.* Anti-thymocyte globulin induces neoangiogenesis and preserves cardiac function after experimental myocardial infarction. *PLoS One* **7**, e52101 (2012).
5. Kong, D.H., Kim, Y.K., Kim, M.R., Jang, J.H. & Lee, S. Emerging Roles of Vascular Cell Adhesion Molecule-1 (VCAM-1) in Immunological Disorders and Cancer. *Int J Mol Sci* **19**(2018).
6. Capuani, S., Malgir, G., Chua, C.Y.X. & Grattoni, A. Advanced strategies to thwart foreign body response to implantable devices. *Bioengineering & Translational Medicine* **n/a**, e10300 (2022).
7. Iatridis, J.C., Wu, J., Yandow, J.A. & Langevin, H.M. Subcutaneous tissue mechanical behavior is linear and viscoelastic under uniaxial tension. *Connect Tissue Res* **44**, 208-217 (2003).
8. Paez-Mayorga, J., *et al.* Neovascularized implantable cell homing encapsulation platform with tunable local immunosuppressant delivery for allogeneic cell transplantation. *Biomaterials* **257**, 120232 (2020).
9. Farina, M., *et al.* Transcutaneously refillable, 3D-printed biopolymeric encapsulation system for the transplantation of endocrine cells. *Biomaterials* **177**, 125-138 (2018).
10. Shapiro, A.M.J., *et al.* Insulin expression and C-peptide in type 1 diabetes subjects implanted with stem cell-derived pancreatic endoderm cells in an encapsulation device. *Cell Reports Medicine* **2**, 100466 (2021).
11. Wang, L.H., *et al.* A bioinspired scaffold for rapid oxygenation of cell encapsulation systems. *Nat Commun* **12**, 5846 (2021).
12. Song, W., *et al.* Engineering transferrable microvascular meshes for subcutaneous islet transplantation. *Nat Commun* **10**, 4602 (2019).
13. Loh, K., *et al.* Inhibition of Y1 receptor signaling improves islet transplant outcome. *Nat Commun* **8**, 490 (2017).
14. Tripathi, D., *et al.* A TLR9 agonist promotes IL-22-dependent pancreatic islet allograft survival in type 1 diabetic mice. *Nat Commun* **7**, 13896 (2016).
15. Marliss, E.B., Nakhoda, A.F., Poussier, P. & Sima, A.A. The diabetic syndrome of the 'BB' Wistar rat: possible relevance to type 1 (insulin-dependent) diabetes in man. *Diabetologia* **22**, 225-232 (1982).
16. Kaino, Y., Ito, T., Goto, Y., Hirai, H. & Kida, K. LACK OF RECURRENCE OF INSULIN-DEPENDENT DIABETES MELLITUS IN SYNGENEIC AND ALLOGENEIC ISLET-TRANSPLANTED DIABETIC BIOBREEDING RATS¹. *Transplantation* **65**, 1543-1548 (1998).
17. Lenzen, S., Arndt, T., Elsner, M., Wedekind, D. & Jörns, A. Rat Models of Human Type 1 Diabetes. *Methods Mol Biol* **2128**, 69-85 (2020).
18. Chen, Y.-G., Mathews, C.E. & Driver, J.P. The Role of NOD Mice in Type 1 Diabetes Research: Lessons from the Past and Recommendations for the Future. *Frontiers in endocrinology* **9**, 51-51 (2018).
19. Coppieters, K.T., *et al.* Demonstration of islet-autoreactive CD8 T cells in insulinitic lesions from recent onset and long-term type 1 diabetes patients. *J Exp Med* **209**, 51-60 (2012).
20. Roep, B.O., Buckner, J., Sawcer, S., Toes, R. & Zipp, F. The problems and promises of research into human immunology and autoimmune disease. *Nat Med* **18**, 48-53 (2012).
21. Takeda, Y., Shimomura, T., Asao, H. & Wakabayashi, I. Relationship between Immunological Abnormalities in Rat Models of Diabetes Mellitus and the Amplification Circuits for Diabetes. *Journal of diabetes research* **2017**, 4275851-4275851 (2017).

22. Pandey, S. & Dvorakova, M.C. Future Perspective of Diabetic Animal Models. *Endocr Metab Immune Disord Drug Targets* **20**, 25-38 (2020).
23. Burke, J.A., *et al.* Subcutaneous nanotherapy repurposes the immunosuppressive mechanism of rapamycin to enhance allogeneic islet graft viability. *Nature Nanotechnology* (2022).
24. Vlahos, A.E., Talior-Volodarsky, I., Kinney, S.M. & Sefton, M.V. A scalable device-less biomaterial approach for subcutaneous islet transplantation. *Biomaterials* **269**, 120499 (2021).
25. Smink, A.M., *et al.* Stimulation of vascularization of a subcutaneous scaffold applicable for pancreatic islet-transplantation enhances immediate post-transplant islet graft function but not long-term normoglycemia. *Journal of Biomedical Materials Research Part A* **105**, 2533-2542 (2017).
26. Carnicer-Lombarte, A., Chen, S.T., Malliaras, G.G. & Barone, D.G. Foreign Body Reaction to Implanted Biomaterials and Its Impact in Nerve Neuroprosthetics. *Front Bioeng Biotechnol* **9**, 622524 (2021).
27. Paez-Mayorga, J., *et al.* Enhanced In Vivo Vascularization of 3D-Printed Cell Encapsulation Device Using Platelet-Rich Plasma and Mesenchymal Stem Cells. *Adv Healthc Mater* **9**, e2000670 (2020).

REVIEWER COMMENTS

Reviewer #2 (Remarks to the Author):

I appreciate the revision and the additional work the authors have done. I went through the manuscript and had two more comments:

1. The NHP experiment is the most impactful in this work. However, not very much information was given except one image with one islet (which didn't look intact or perfectly healthy). It would be helpful to provide more information as what the cell viability was in terms of both the overall islet population and individual islet for each NHP. I understand this may be difficult to determine precisely but any additional images/data and estimate would be helpful.

2. In the end of the Abstract, the authors stated "In sum, the NICHE is a safe and effective platform for islet transplantation and long-term T1D management." This is a strong statement and I suggest the authors tone down a little it to avoid confusion and misunderstanding from audience.

We thank the reviewer for their interest and insightful comments on our work. Here, we present a point-by-point response to the reviewer's comments.

Reviewer #2 comments:

1) The NHP experiment is the most impactful in this work. However, not very much information was given except one image with one islet (which didn't look intact or perfectly healthy). It would be helpful to provide more information as what the cell viability was in terms of both the overall islet population and individual islet for each NHP. I understand this may be difficult to determine precisely but any additional images/data and estimate would be helpful.

We appreciate the reviewer's understanding regarding the technical difficulties that hinder precise quantification of islet viability in our NHP study. As islets disperse throughout the cell reservoir, reliable quantification of the overall islet population in NICHE would require homogenization of the entire cell reservoir tissue. However, to maximize our characterization efforts, NICHE tissue was divided for CyTOF and histology assessment, preventing reliable quantification of overall islet viability. In terms of viability at the individual islet level, below we include supplemental micrographs that provide a broader qualitative picture of the NICHE microenvironment with and without IS. NICHE without immunosuppression showed limited islet-like structures with few-to-none insulin-positive cells that were densely infiltrated by inflammatory cells, suggestive of rejection with beta-cell destruction (figure 1a,c). In contrast, islet structures within IS NICHE had multiple insulin-positive stained cells that were distributed in a typical islet pattern. Furthermore, compared to NICHE without IS, there were significantly fewer, mostly peripheral, inflammatory infiltrates (figure 1b, d).

Figure 1. Micrographs of NICHE tissue 14 days post-transplant in NHPs. H&E staining of a) no IS and b) IS NICHE. Corresponding micrographs of c) no IS and d) IS NICHE tissue stained for insulin. Each set of H&E and insulin micrographs corresponds to individual NHPs.

2) In the end of the Abstract, the authors stated "In sum, the NICHE is a safe and effective platform for islet transplantation and long-term T1D management." This is a

strong statement and I suggest the authors tone down a little it to avoid confusion and misunderstanding from audience.

We appreciate the reviewer's suggestion. The sentence has been modified.

REVIEWERS' COMMENTS

Reviewer #2 (Remarks to the Author):

The authors did a good job and I have no more comments